

# Numerical investigation of the Arctic ice-ocean boundary layer; implications for air-sea gas fluxes

A. Bigdeli[1], B. Loose[1], S. T. Cole[2]

[1]Graduate School of Oceanography, University of Rhode Island, Rhode Island, 02882, U.S.A
[2]Woods Hole Oceanographic Institution, Woods Hole, Massachusetts, 02543, U.S.A

*Correspondence to*: A.Bigdeli (Arash_Bigdeli@uri.edu)

**Abstract.** In ice-covered regions it can be challenging to determine air-sea exchange – for heat and momentum, but also for gases like carbon dioxide and methane. The harsh environment and relative data scarcity make it difficult to characterize even the physical properties of the ocean surface. Here, we seek a mechanistic interpretation for the rate of air-sea gas exchange (k) derived from radon-deficits. These require an estimate of the water column history extending 30 days prior to sampling. We used coarse resolution (36km) regional configuration of the MITgcm with fine near surface vertical spacing (2m) to evaluate the capability of the model to reproduce conditions prior to sampling. The model is used to estimate sea-ice velocity, concentration and mixed-layer depth experienced by the water column. We then compared the model results to existing field data including satellite, moorings and Ice-tethered profilers. We found that model-derived sea-ice coverage is 88 to 98% accurate averaged over Beaufort Gyre, sea-ice velocities have 78% correlation which resulted in 2 km/day error in 30 day trajectory of sea-ice. The model demonstrated the capacity to capture the broad trends in the mixed layer although with a bias and model water velocities showed only 29% correlation with actual data. Overall, we find the course resolution model to be an inadequate surrogate for sparse data, however the simulation results are a slight improvement over several of the simplifying assumptions that are often made when surface ocean geochemistry, including the use of a constant mixed layer depth and a velocity profile that is purely wind-driven.

## 1 Introduction

The ocean surface is a dynamic region where momentum, heat and salt, as well as biogeochemical compounds are exchanged with the atmosphere and with the deep ocean. These fluxes drive vertical mixing that generates the mixed layer. On the top of the mixed layer, at the sea-air interface, gases of biogenic origin and geochemical significance are exchanged with the atmosphere. Theory indicates that the aqueous viscous sublayer, which has a length scale of 20 to 200 μm, (Jähne and Haubecker 1998) is the primary bottleneck for air-water exchange. Limitations in measurement at this critical scale have led to approximations of sea-air gas exchange based on indirect measurements. Four approaches involving data are typically used (Bender et al. 2011), 1) Parametrization of the turbulent kinetic energy (TKE) at the base of the viscous sublayer 2) Tracing purposefully injected gases (Ho et al. 2006; Nightingale et al. 2000) 3) Micro Meteorological methods (H. J. Zemmelink et al. 2006, 2008; Blomquist et al. 2010; Salter et al. 2011), and 4) Radon-deficit Method. Here, we examine the





radon-deficit method (4), together with a parameterization of the TKE forcing (1) that theoretically leads to the observed deficit in mixed-layer radon.

TKE is mostly dominated by wind speed on open ocean interface (Wanninkhof 1992; Ho et al. 2006; Wanninkhof and McGillis 1999; Nightingale et al. 2000; Sweeney et al. 2007; Takahashi et al. 2009). In the polar oceans wind energy and
atmospheric forcing are transferred in a more complex manner as a result of sea ice cover (Loose et al. 2009, 2014; Legge et al. 2015). Sea ice drift due to Ekman flow (McPhee and Martinson 1992), freezing and melting of ice leads on the surface ocean (Morison et al. 1992) and short period waves (Wadhams et al. 1986; Kohout and Meylan 2008) all constitute important sources of momentum transfer. Considering the scarcity of data on marginally covered sea-ice zones (Johnson et al. 2007; Gerdes and KöBerle 2007), especially during arctic winter time, comparing and validating these models are quite
challenging.

The radon deficit method involves sampling 222Rn and 226Ra in the mixed layer to examine any difference in the concentration or (radio) activity of the two species. Radon is a gas, radium is a cation; in absence of gas exchange 222Rn and 226Ra enter secular equilibrium meaning the amount of 222Rn produced is equal to decay rate of 226Ra. Any missing 222Rn in the mixed layer is attributed to exchange with atmosphere (Peng et al. 1979).
Since the 222Rn concentration in air is very low, less than 5% (Smethie et al. 1985) and considering that concentration is proportional to activity/decay rate A, we can use Eq. (1) to determine gas exchange. Where k gas transfer velocity in (m d-1), AE is the activity or decay rate of 222Rn which in secular equilibrium is equal to 226Ra activity, AM is 222Rn measured decay rate in mixed layer, $\lambda$ is decay constant of 222Rn (0.181 d-1) and h is the mixed layer depth

$$k = [\frac{A_E}{A_M} - 1]\lambda h , \qquad (1)$$

Gas transfer velocities from equation (1) reflect the memory of 222Rn for a period of two to four weeks (Bender et al. 2011), which is four to eight times the half-life of 222Rn (3.8 days).The mixed layer depth, h, is calculated from the measurements performed at the hydrographic stations during 222Rn sampling process.

While a valuable tool this method is based on two premises, a) an invariant mixed layer and b) invariant TKE forcing during the entire period of "memory" that gas in the mixed-layer has experienced. To further illustrate the discrepancy caused by
utilizing the invariant assumptions, described above, consider a mixed layer that rapidly changes by a factor of 2 just prior to sampling for radon. If the mixed-layer becomes shallower (stratification) h – defined by the density profile - will be smaller by factor of 0.5 while AE/AM in the mixed layer remains the same. Based on equation 1, this causes k to be half of its true value. That is, prior to stratification TKE forcing was sufficient to ventilate the ocean to a depth greater than the apparent h (Bender et al. 2011).
Conversely, if the mixed layer deepens due to mixing, h increases and a new parcel of water with AE/AM= 1 is added to mixed layer, causing the activity ratio to come closer to unity. These two influences on equation 1 (increasing H and AE/AM approaching unity) work against each other, but the net effect is to cause k to appear larger. The change of factor of 2 in



mixed layer depth in less than two weeks has been observed during several studies (Acreman and Jeffery 2007; Ohno et al. 2008; Kara 2003).

The "memory" of gas exchange forcing that radon experiences is further complicated by the presence of sea ice. Consider two alternate water parcel drift paths that lead to the 222Rn sampling station in sea-ice zone (Fig. 1). Path B demonstrates a

history which water column spends most its back trajectory under sea-ice. Path A shows a water column which experiences stratification and shallowing of mixed layer depth equal to δh when drifting through water that is completely uncovered by ice. During most of Path B gas transfer happens in form of diffusion through sea-ice and it will have a very low k (Crabeck et al. 2014; Loose et al. 2011). In contrast Path A will have a greater radon deficit, but a smaller h because of stratification. In either case, it is critical to take into account the time history of gas exchange forcing, including changes in the mixed-layer

and ice cover, which has led to the apparent radon deficit at the time of measurement.

This observation about drift paths in the sea ice zone strongly implies that we must consider both time and space in estimating the forcing conditions that are recorded in the radon deficit. In other words, we require a Lagrangian back trajectory of water parcels to track the evolution of mixed layer and its relative velocity 4 weeks prior to sampling.

Although satellite data, Ice tethered drifters (Krishfield et al. 2008) and moorings (Krishfield et al. 2014; Proshutinsky et al.

2009) have provided valuable seasonal and spatial information about the sea ice zone, they do not track individual water parcels and tend to convolve space and time variations. The spatial limitation of these data pose a challenge to producing a a back trajectory of the water parcel hence we used a 3D numerical model to simulate these data.

This is the principle goal of our study – we want to know if a coarse resolution model that can be run on a desktop multi-core processor can provide us with spatial and temporal information to better model the back trajectory of a radon-labelled water

parcel. The null hypothesis might therefore be, that simple assumptions such as an invariant mixed-layer, Ekman ice drift and area-averaging of sea ice cover are as good or better at reproducing the radon-based estimates of k.

We set up a numerical simulation that captures near surface phenomena as a means to access the history of a water parcel sampled for radon, and the forcing conditions that led to the observed radon deficit. For this study we are focusing on near surface phenomena in the Arctic using the Arctic Regional configuration of the MITgcm with the domain of ECCO2

(Marshall et al. 1997; Menemenlis et al. 2008; Losch et al. 2010; Heimbach et al. 2010; Nguyen et al. 2011). A number of Arctic ocean-ice models which have been compared as part of Arctic Ocean Models Intercomparison Project (AOMIP) (Proshutinsky et al. 2001; Lindsay and Rothrock 1995; Proshutinsky et al. 2008) in regard to their capability to represent main ice-ocean dynamics. This model has shown to have better correlation with remotely sensed sea ice data (Johnson et al. 2012) and high near surface vertical resolution.

It is worth noting that model studies using higher resolution evaluate model skill are already in print.  For example, Holloway et al., (2011) use a 4 (km) version of the MITgcm regional model has a high skill in producing near surface velocities and capturing eddies. So, the choice to use the 36 (km) resolution model may seem anachronistic. However, we approach this problem as geochemists who are searching for a data interpretation tool. We seek higher resolution in the top





100 m of the water column, as compared to these prior studies. Furthermore, the 36 (km) model can be run on a desktop multi-core processor, and the simulation is completed in less than a week.

The remainder of the article is organized as follows: In section 2 we introduce the changes we made to increase the near surface resolution. Section 3.1 and 3.2 contain the comparison of the new results and original model outputs including sea-

ice concentration, velocity and trajectory to observed data from satellite and Ice tethered profiler. Sections 3.3, 3.4 and 3.5 are used to study the model output salinity and temperature structure resulting upper ocean density structure and mixed layer. Sections 3.6 and 3.7 evaluate the qualitative model circulation and correlation in near surface water velocity. Lastly, section 4 shows the summary of our results. In each section, we attempt to compare model output to direct observations, and to simple assumptions – such as those that are commonly made in geochemical studies of the upper ocean. These comparisons

help define the relative value of the model output to geochemical data interpretation.

## 2 Method

The model has a horizontal resolution of 36km, and 50 vertical layers employing the z* coordinate system (Adcroft and Campin 2004) and the maximum model depth of 6150m. The surface forcing is sourced from ERA40 and ECMWF numerical weather forecast, and the boundary conditions are taken from output of the ECCO2 global model (Menemenlis et

al. 2008) and 25 year Japanese Reanalysis Project (JRA25)(Onogi et al. 2007). Initial conditions are from World Ocean Atlas 2005 (Antonov et al.; Locarnini et al.) and initial sea ice conditions are from (Zhang and Rothrock 2003), model is allowed to spin up from 1979 to 1992. The vertical mixing uses K profile parameterization (KPP) developed by Large et al. (1994) and salt plume parameterization of Nguyen et al. (2009).

We introduced a new vertical grid spacing (hereafter referred to as A1), 2 meter steps near surface which continue to the

depth of 50 meter which then gradually increase to a maximum interval of 650 meters in deep ocean contrasting to the optimized model (called A0) which had 10 meter intervals near surface, progressively increasing to 450 meter at maximum depth. All the boundary conditions from ECCO2 have been interpolated to match the new grid system. Due to stability condition of the model we also reduced the time steps of the model from 2400 seconds to 600 seconds. The model simulates the time period from January 1992 to September 2013. This new grid system allows us to capture near surface small details

which cannot be represented with the coarser grid system.

Satellite estimation of sea-ice cover at 25km horizontal resolution (Comiso 2000) is being interpolated on a grid system and then compared with A0 and A1. Then sea-ice drift gathered by 28 Ice Tethered Profilers (ITP) (Krishfield et al. 2008) which has more than 2 month of data in Beaufort Sea between 2006 and 2013 has been used to do the ice velocity and trajectory comparison. We selected only ITPs with 2 or more months of drift data in the Beaufort Sea.

We compared near surface water velocity data from ITP-V (Williams et al. 2010) and upward looking Acoustic Doppler Current Profiler installed on McLane Moored Profiler (MMP) (Proshutinsky et al. 2009; Krishfield et al. 2014) to A1 in order to compute the accuracy and feasibility of calculating back trajectory of parcels located at the mixed layer.



Using salinity and temperature profiles from ITPs (Krishfield et al. 2008) we calculated mixed layer depth and the average geopotential height between 2006 and 2013 and compared it to 2m vertical resolution model output(A1).

## 3 Results

### 3.1 Sea ice concentration

Both A0 and A1 model scenarios are used to study sea ice extent (area of cells with more than 15% ice cover) from 1992-2013. At the Arctic Basin scale, the model depicts a decrease in the September minimum sea ice extent of 0.82 million square kilometres per decade or 10% of the starting value at 1992 compared to satellite data which display a 1.2 million square kilometres per decade, the model show an over estimation of sea-ice extent (Fig. 2). For further analysis we introduced a grid system covering the Beaufort Gyre and interpolated the data from satellite (Comiso 2008) and A0 and A1

on to the grid.

The analysis grid extends from 70° to 80° north and 130° to 170° west, covering most of Beaufort Gyre (Fig. 3). Grid points can be divided into two main geographic zones that are marked out based on sea ice cover. The first zone contains grid points where the annual average sea ice cover is greater than 80%. These sets of points are fully covered by sea-ice most of the year. The second zone can be described as "marginally ice covered" wherein the ocean surface is free of ice for some

15 fraction of the year. We chose 3 points within this sea ice geography to compare the seasonal and interannual behavior of the model with satellite ice cover. The points are located at 80°,131.82° (P1), 70.82° 169.82° (P2), and 74.76° north and 163.51° west (P3).

The ice cover at P1, P2 and P3 (Fig. 4) can be divided into 3 ice phases: (a) Fully covered in ice, (b) Open water and (c) a transition between (a) and (b). P3, which is the furthest south, has all three phases. In contrast P1 ice cover only dips below

20 60% for two brief periods during the 7 year time series depicted in Fig 4 - once in 2008 and again in 2012. These three points illustrate where and when the model has the greatest challenge reproducing the actual sea ice cover. At the extremities of the ice pack, where the water is predominantly covered by 100% or 0% ice (P1 and P3), the model captures the seasonal advance and retreat and the percent ice cover itself is accurate. However, in the transition regions that are characterized by marginal ice for much of the year (P2), the model has more difficulty reproducing accurate sea ice cover as well as the

25 timing of the advance and retreat. These results are the same for both A0 and A1 experiments, and this behavior is consistent with the description that has been explained by Johnson et al. (2007), that models have a higher accuracy predicting sea ice concentration in central arctic and less accuracy near periphery and lower latitudes.

The spatial sensitivity of the model can be observed using root mean square (RMS) error Eq. (2), calculated over the 1992-2013 period (Fig. 5). The area with most error coincides with area between the 80% and 60% contour lines (Fig. 3) and is

30 concentrated primarily in the Western Beaufort. The RMSE error of 0.2 is the maximum value away from land, this same level of error can also be found near land which is caused by fast-ice generation. Fast Ice in the model is replaced with pack





of drifting sea ice; this error is common between numerical models and has been studied during AOMIP (Johnson et al. 2012, p. 20).

$$RMSE(point) = \sqrt{\sum_{i=1}^{n}(C_{simulation} - C_{satellite})^2 /n} \,, \qquad (2)$$

If we compare the monthly climatology for sea ice cover over the 1992-2013 period, the RMS error between model and
satellite data is least during the early winter months (e.g. Jan-Mar) when sea ice is close to its maximum extent. Comparing A0 and A1 Fig. 5 depicts an increase in RMSE during July, August, September and October and a minor decrease in May and November. The RMSE appears to be greater during the summer months of ice retreat, and slightly less during the autumn months of ice advance. Overall, the periods of transition (melt and freeze) coincide with the greatest RMSE.

The 2m revised grid (A1), with smaller vertical intervals near the surface has produced a greater RMSE than the optimized
model (A0). We are still exploring why this took place. It is possible that the convective parameterization in the ice model is somehow negatively affected by short i.e. 2 m vertical layers. To reproduce changes in the mixed layer will require the greater resolution reflected in the A1 model run.

**3.2 Sea ice velocity and trajectory**

Ekman turning causes ice and water to move at divergent angles with respect to each other. Ice moves the fastest, with mean
values of 0.09 m s$^{-1}$ (Cole et al. 2014), and the water column progressively winds down in velocity, along the Ekman spiral. Stratification in the Arctic leads to a confinement of the shear stress closer to the air-sea interface and also produces greater divergent flow vectors between ice and water (McPhee 2012). In the marginal ice zone or in regions where ice is converging or diverging, these motions, relative to the motion of the water column can produce significant changes in the water column momentum budget as well as air-sea fluxes. Thankfully, the ITPs can provide us with a measure of the real ice drift.
For purposes of comparison with the model and ITP drifts estimated a simple sea-ice velocity vector field Eq. (3) using reanalysis wind data (Onogi et al. 2007), with "a" and θ as empirical coefficients equal to 0.019 and 28 degree (Cole et al. 2014), This simple estimation neglects feedback from Ekman spiral under the ice which has been implemented in more sophisticated analytical solutions (Park and Stewart 2015).

$$\boldsymbol{u}_{ice} = \mathrm{a}\boldsymbol{u}\mathrm{e}^{-i\theta} \,, \qquad (3)$$

To generate a more quantitative comparison between the results we utilized the same method used by Timmermans et al. (2011), comparing ice velocity components (u-v) of A0, A1 and estimated velocity from Eq. (3) to ITP velocity and finding the correlation coefficient of each experiment versus the daily averaged actual drift velocity of the ITPs (Fig. 6)
By averaging over all the ITPs operating in Beaufort Gyre during 2006 to 2013, A0 had 0.78, A1 0.77 and simple estimation had 0.65 correlation with actual velocity components.





As stated in the introduction, one of the principle objectives of this study is to evaluate the capability of the MITgcm to produce useful forcing field on the time-scale of mixed layer memory for biological and geochemical compounds, particularly the volatile ones. For a trace gas such as oxygen or radon, we only require adequate representation of the time-history of hydrographic conditions that act upon on a tracer for ~30 days, because this is the approximate renewal time of the

mixed layer, based on a representative value of air-sea exchange. Consequently, we need the ice that covers our Lagrangian water parcel, so we calculated 30 day trajectories of sea-ice. We generated the ice floe trajectories by step-wise integration of the daily-average ice model output of velocity and direction in order to follow the individual ice floe. To illustrate, we have selected one of the 28 ITPs - ITP53 during its operation in Beaufort Sea from August 6, 2011 to August 13, 2012.

To generate a statistically large sample size, this calculation is preformed such that every day is treated as a starting point for

simulation and the trajectory of that parcel is traced for a 30 day period. Figure 7 shows the result of this calculation for ITP 53. Taking the separation between the actual ITP trajectory after 30 days and the model trajectory, we determine the average drift error. Using 28 ITPs and 11744 operation days, for a total of N = 352320 data points. The average separation is 59.3 (km) for A0 and 62.5 (km) for A1 that is equal to 1.97 and 2.08 (km/day) error respectively. This amounts to a 5% increase in the error from A0 to A1. Same calculation for trajectories based on Ekman drift sea-ice velocities computed from Eq. (3)

yield 4.3 km/day error. In other words, the coarse resolution model error is approximately half as large as if we assumed a simple Ekman drift for the ice trajectory.

The path of the ice, seen through its trajectory, appears to show a dependence of the choice of resolution in the vertical grid cell, but we cannot say for certain if it improves or deteriorates the results. ITP-52 has correlation coefficient of 0.87 on northward direction and 0.82 on eastward direction but due to accumulation of error both A0 and A1 fail to represent the

actual trajectory of ITP-52 using just the start point and letting error accumulate through the entire ITP path that is 374 results in a pronounced difference in end points (Fig. 10).

### 3.1 Temperature and salinity, and Geopotential height

### 3.3.1 Geopotential height

McPhee (2013) showed that the accumulation of freshwater in the Beaufort Gyre led to an increase in geopotential height

and a strong geostrophic flow that may have feedbacks for the advection of ice out of the Canadian Arctic. Here, we have used the model to calculate geopotential height (Fig. 12), referenced to the 400-m isobath and averaged from 2008-2011, with velocity vectors calculated based on the thermal wind equation. We can compare our result to hydrographic reports generated from ITPs and CTD cast during the same time period which has been reported in previous literature (McPhee 2013).

Hydrographic data show a doming of about 45+ (cm) relative to geoid in Beaufort Gyre with the maximum centered at 74.5° N and 150° W (Kwok and Morison 2011, p. 20). Our model results show the same dome and correct sign, producing the correct geostrophic flow direction with the magnitude of 100+ (cm). The model shows the center of the dome to be around



80° N and 140W. Discrepancy between the magnitudes is in line with the results from sea-ice concentration section, with model over predicting the sea-ice extent hence generating excess fresh water. In a big picture this shows the capability of the model to predict the freshening of top 400 meter.

### 3.3.1 Vertical Salinity Temperature profiles

We chose 4 profiles in Beaufort sea to represent the simulated vertical salinity and temperature. The first two sets of profiles are from ITP-1 winter (Fig. 11) and summer 2006 (Fig. 11). The third set is from ITP-43 during winter 2010 (Fig. 12) and the fourth is from ITP-13 during summer 2008 (Fig. 13).

During winter time, a well mixed layer reaches below 15 meters in the model T and S profiles (Fig. 11), followed by a very large gradient. The mixed-layer temperature is close to the local freezing point in a condition called "ice bath" (Shaw et al.
2009). The ITP profiles are similar; however the ITP mixed layer depth is deeper by nearly 10 meters, indicating more ice formation and convective heat loss over this water column, as compared to the model water column. In summer (Fig. 11) the model mixed layer shoals to approximately 5 meters depth following two local temperature extrema, the bigger maximum is at ~35 meters generated by intrusion of Pacific Summer Water (PSW) which is a dominant feature in Canada basin. The second smaller maximum happens around 10 meters called Summer Mixed Layer (Shimada et al. 2001) or Near-Surface
Temperature Maximum (NSTM) (Jackson et al. 2010) which is a seasonal feature generated by shortwave solar heat diffusion (Donald K. Perovich 1990). These two well-defined phenomena are broadly descriptive of the summer surface layer in the Beaufort Gyre. They are; however, absent from the ITP data at this location, indicating a different ice and heat budget time history. The mismatch in temperature is one of the challenges that numerical models face when reproducing the cold halocline. Recent studies show that eddies with diameter of approximately 30 km (Nguyen et al. 2012) have a major
contribution in transport of PW from shelf break into Canadian basin, which requires a model with finer horizontal resolution.

Data and model profiles in Fig. 12 show better agreement in the shape and the absolute value of the T and S profiles. Both model and ITP data have a 20 meter deep mixed layer during 2010 winter. The model in this case does not show as much change in vertical temperature structure compared to actual data. In the profile from ITP-13 (Fig. 13) the model again over
estimates the temperature beneath the mixed layer, although certain features including the NSTM can still be found near 10 meters, yet not as pronounced since it is very close to PSW. Bearing in mind that density in the Arctic is dominated by changes in salinity, we move forward to density profiles from this point on.

### 3.4 Density profiles

We compare the MITgcm density to the time series of density profiles from ITP-62 (Fig. 15-16) during the course of nearly a
30   year starting in Sep 2012. A black mask indicates locations where there is no data from ITP-62 - particularly in the upper 7 meters of the water column. ITP-62 transited through Canadian basin, density profiles have both temporal and spatial changes in them (Fig. 14).



We are able to discern some broad similarities in the model and ITP density profiles. From September to mid-November, the density profile above 50 (m) tended to increase, consistent with the period of cooling and ice formation. From December through March, both ITP and model density profiles remain relatively constant. Between March and April, ITP-62 appears to drift through a unique water parcel, with lower density above 70 (m). The same feature can be observed in the MITgcm density. However, on a smaller scale, there is significantly more variation in the ITP data than what the model represents.

For exploring the reason behind the density signals we are going to use the simulated fraction of sea ice cover and ice thickness (Fig. 17). The dominating effect appears to result from sea ice fraction and when there is continuously covered area. The changes from sea ice thickness can be observed in volume of fresh water in the water column. A peak in near surface density can be seen late in March when a decrease in ice fraction from 100% to 90% exposed the surface water to cold atmosphere, which generated newly formed sea ice and inserted brine into the water column. This signal will be further discussed on mixed-layer section.

## 3.5 Mixed layer depth

There are many different methods in the literature for calculating mixed layer depth (Brainerd and Gregg 1995; Wijesekera and Gregg 1996; Thomson and Fine 2003; de Boyer Montégut et al. 2004; Lorbacher et al. 2006; Shaw et al. 2009). The methods can be divided into two main types (Dong et al. 2008): The first type of algorithm looks for the depth ($z_{MLD}$) at which there has been a density increase of $\delta\rho$ between the ocean surface and $z_{MLD}$. A typical range of values for $\delta\rho$ are 0.005 (kg m$^{-3}$) to 0.125 (kg m$^{-3}$) (Brainerd and Gregg 1995; de Boyer Montégut et al. 2004). The second type uses slightly different criteria, where the base of the mixed layer is determined as the depth where the gradient of density ($\partial\rho/\partial z$) equals or exceeds a threshold; typical numbers for ($\partial\rho/\partial z$) are 0.005 (kg m$^{-4}$) to 0.05 (kg m$^{-4}$) (Brainerd and Gregg 1995; Lorbacher et al. 2006). A more sophisticated approach to type 1 of this criteria is to utilize a differential between ($\rho_{100m}$ - $\rho_{surface}$) as the cut of point (instead of using a fixed $\delta\rho$) to account for the effects of surface $\rho$ changes during winter and summer (Shaw et al. 2009). Here, we have implemented two of these methods M1 and M2, with M1 using $\delta\rho$ equal to 0.2 of ($\rho_{100m}-\rho_{surface}$) (Shaw et al. 2009) and M2 with a gradient ($\partial\rho/\partial z$) cut off point equal to 0.02 (kg m$^{-4}$) which matches innate model parametrization of MLD (Nguyen et al. 2009).

We compare these 2 methods by applying them to the profiles from Fig. (10-11-12-13) which result in (Fig. 18). In case (a) and (b) M1 produces a mixed layer depth that is 8 to 12 meters deeper, compared to the other method. A visual examination of profiles appears to indicate that the M1 criteria may be too flexible of a criteria. The results from M2 appear to be intermittently "realistic", whereas M1 can be difficult to implement on high resolution data with greater small-scale variability. In practice, we find M2 is the most straight-forward to implement.

It should be mentioned that it is difficult to consistently compare performance of the M1($\delta\rho$) and M2 methods on ITP and model data, because the model data extends to the free surface, but the ITP data stops at 7 (m) depth and it has been shown that summer mixed layer in the Canada basin can be less than 12 meters (Toole et al. 2010). To account for this effect, we apply an additional restriction wherein any profile whose mixed-layer depth is less than 2 (m) below the shallowest ITP



measurement is discarded. This restriction effectively removes any ML depths shallower than 10 meters due to ITP sampler not resolving the upper 8 meters of water column. In some cases, a remnant mixed-layer from the previous winter may exist in the water column. In this case, the methods incorrectly identify the remnant ML as actual ML depth.

To compare the methods over a longer time period, we calculated the mixed-layer depth from model data and ITP-62 data along the ITP-62 drift track. We used both M1 and M2 to determine the ML depth for the model data and for ITP-62 data (Fig. 19-20). M1 and M2 show almost similar results. The model shows a shallower ML compared to the ITP data; the most prominent feature in early March corresponds to a sudden change in density found in (Fig. 15). During the months of June and July, the model predicts zero mixed layers with stratification almost to the surface. The ITP data, beginning at 8 m cannot reflect this stratification, but we know this model result to be plausible based upon our comparisons with shipboard CTD profiles (data not shown here) and what is known about ice melting and stratification.

For further exploring the forcing that drives the mixed layer, we use the Japanese reanalysis (JRA-25) (Onogi et al. 2007) wind speed, air temperature data and sea-ice fraction from the model and interpolated them on the path of ITP-62. The prominent feature during March to April can be explained by the reduction in ice cover to 95% , which resulted in water column exposure to low or moderate wind from 5 $(ms^{-1})$ to 10 $(ms^{-1})$ and low air temperatures, which in turn increased mixing and deepening of ML by 100% (Fig. 17). The wind appears to have led to a divergence in ice cover, which in-turn exposed the ocean to the cold atmosphere, leading to a loss of buoyancy and an increase in the mixed-layer depth. In this regard, the wind appears to play a facilitating role, leading to sea ice divergence, but there is no evidence to support that a strong wind event caused the mixed layer to deepen.

Further exploring the dependency of ML evolution to wind and temperature we calculated cross correlation between ML-wind and ML-temperature (Fig. 22). This analysis shows responses with respect to air temperature happen with no significant lag and wind with 10 days of delay which would result in a maximum 0.36 correlation for wind and 0.39 correlation for temperature. Without accounting for the effect of sea ice cover, which can somewhat modulate momentum inputs by the wind and strongly modulate heat and moisture fluxes, it appears that neither temperature nor wind has a dominant effect on MLD in the Arctic, but rather both play a role.

## 3.6 Circulation

The time-averaged upper ocean circulation in the Arctic has been described by oceanographers based upon the origin of the water masses that enter and exit the Arctic Ocean. These are primarily the Atlantic water (AW) and Pacific Water (PW). After entering the Arctic ocean AW and PW loose heat and sink, AW to between 200 (m) and 800 (m) (Golubeva and Platov 2007) and PW to between 50 and 150m (Steele et al. 2004). During the AOMIP intercomparison study it was observed that Arctic models can produce two AW circulation in opposing directions, with observations suggesting a cyclonic circulation in Arctic basins (Lindsay and Rothrock 1995). Potential vorticity influx from sub-Arctic oceans (Yang 2005) and unresolved eddy parametrization (Holloway et al. 2007) have been shown to diminish this discrepancy and create consistent cyclonic circulation among all models (Proshutinsky et al. 2011).





We evaluated the capability of the model to reproduce the general circulation of Arctic by averaging output ocean velocities from A1 experiment during a time span of seven years from 2006 to 2012. We focused on two depth intervals 1-15 (m) depth to represent Ekman layer (Cole et al. 2014) (Fig. 24) and the second set is 180-250 m to model AW circulation (Fig. 23).

The flow field (Fig. 23) representing AW displays a counter-clockwise topographic boundary current in the Beaufort and Chukchi seas matching AW cyclonic circulation (Lindsay and Rothrock 1995; Proshutinsky et al. 2011). In contrast there is a clockwise boundary current in the southern Canadian basin (Fig. 24). These circulation patterns qualitatively match the hydrographic description of the time-mean circulation in the Arctic.

### 3.7 Velocities in the water column beneath drifting sea ice

We have very little information from direct observations that permit us to track a water parcel especially beneath sea ice. This is one area where model output could be critical as there are not obvious alternatives. To judge the consistency of the model water current field, we compared 2D model water velocity to data gathered from two sources: (1) from ADCPs mounted on moorings that were deployed starting in 2008 in Beaufort Gyre (Proshutinsky et al. 2009) and (2) the ITP-V sensor equipped with MAVS (Modular Acoustic Velocity Sensors) (Williams et al. 2010), which was the only operating ITP

before 2013 which had an acoustic sensor mounted on it.

We compared the magnitude of velocity without accounting for flow direction, i.e. removing the effects of Ekman turning (Cole et al. 2014) in order to find a 2D-correlation over the duration of ITP-V working days which is from Oct 9, 2009 to Mar 31, 2010 (Fig. 25). The ITP data has been daily averaged so it matches the time interval of the model structure. Due to ITPs mechanical limitation on reporting velocities shallower than ~6 meters, the comparison starts from 7 meters.

The first result that emerges is that ITP velocity shows a lot more structure, changing speed at a much higher time frequency most probably caused by eddies which has been shown to be a characteristic feature in the top layers of Beaufort Sea (Zhao et al. 2014). Figure 26-27 depicts the simulated and observed velocity components for ITP-V, to better compare the results the data and simulation have been filtered by 4 days low pass filter corresponding to 36 (km) in distance which the ITP travels effectively low pass filtering the results.

The absolute 2D-correlation between the simulated and observed velocities for eastward direction is 20%, northward direction 22% and for the speed is 20%, these numbers represent r and not r2. We also compared the correlation versus depth and correlation of the velocity components and current speed (Fig. 28). If averaged over depth the northward velocities will have about 28% correlation with depth averaged data and also the correlation increases with depth after 40 meters. One possible reason for this lack of correlation near surface is effects of stratification in the model due to its shallower mixed

layer.

Another source of ocean current data from the Arctic are ADCPs mounted on the moorings which has been deployed as part of Beaufort Gyre Observation System (BGOS), these profilers are located on the top float of subsurface moorings, measuring water velocities within the upper 30 meters of water column (Krishfield et al. 2014). Mooring D was the first





mooring outfitted with an ADCP in 2005. Figure 29 shows the current speed measured by this mooring from October 2010, covering 1032 days of data located at 73.99° N and 139.98° W.

Simulation results follow certain features of the actual data (Fig. 29), high velocities around Jan-2011, Oct-2011 and Oct-2012 which is caused by exposure of water column surface to winds. In general, higher velocity magnitudes are observed which are not represented in the simulation.

For further exploring these high velocity features in the mooring data, we used wind speed from Japanese Reanalysis data (Onogi et al. 2007), sea-ice cover from satellite, constant sea-ice draft of 2m, sea-ice velocity estimation based on Sect. 3.2, Eq. (3) to construct a simple Ekman spiral based on Eq. (4) and Eq. (5). Our assumptions include a neutrally buoyant layer, constant dimensionless eddy viscosity ($K^* =0.2$) (McPhee and Martinson 1992; McPhee 2012) and that our target depth is outside of the boundary (log) layer sea ice (Cole et al. 2014).

$$\boldsymbol{\tau}_{(z)} = u_{*0}\boldsymbol{u}_{*0}e^{\left(1/\sqrt{2K_*}\right)(1-i)fz/u_{*0}} , \tag{4}$$

$$\boldsymbol{u} = \mathrm{d}(\boldsymbol{\tau}_{(z)})/z , \tag{5}$$

$$\boldsymbol{u}_{total} = (1 - C_{ice})\boldsymbol{u}_{open\,water} + C_{ice}\boldsymbol{u}_{sea\,ice\,covered}, \tag{6}$$

With z the distance from the water interface and $\boldsymbol{\tau}_{(z)}$ representing the shear vector, f Coriolis parameter and $C_{ice}$ is the fraction of sea-ice cover. The interface friction velocity vector ($\mathbf{u}_{*0}$) is calculated based on the type of interface. In case of water-ice interface $\mathbf{u}_{*0}$ is based on law of the wall (McPhee 2008). Total Ekman velocity is $C_{ice}$ weighted summation of Ekman vectors based on ice and air (Eq. 6), we then compared the velocity components based on this method, simulation and data (Fig. 30-31).

### 3.7.1 Sources of error in water column current velocities

There is 29% correlation between the simulation and mooring data. This is nearly identical to the poor correlation observed between the model and the ITP velocities. These discrepancies between model and data can be caused by several reasons. The first is the model resolution is too coarse to resolve mesoscale and submesoscale eddies, which have characteristic lengths of 10 km or smaller (Boccaletti et al. 2007; Timmermans et al. 2008). Because these eddies would all hypothetically fit within one of our 36-km grid boxes, this model configuration lacks the ability to represent them. Eddy resolving models have better agreement with velocity data (Holloway et al. 2011). Secondly the parameterization of sea-ice in the current model utilizes "levitating" sea-ice (Losch et al. 2010), meaning the sea-ice always stays on top of the free surface. This assumption neglects the forcing caused by sea-ice advection whenever surface velocity is different than the sea-ice velocity (Campin et al. 2008). Neglecting the actual draft of sea-ice also introduces an error in calculating the correct distance from boundary which in turn makes the direction of velocity in Ekman spiral unreliable. Omission of other physics such as tidal waves, which is a common trait between arctic models, may as well have an effect on the velocity fields.





Having established that the correlation between the model output and mooring or ITP data is generally poor (i.e. 29% as reported above), we can further ask whether, in the absence of data, we would be just as well served by assuming that the current speeds are purely Ekman in nature. Using Eq. 4-6, as described above, we generated the Ekman spiral under partial ice cover to measure the current speed using only reanalysis winds. As with the model data, we compute the vertically-averaged current speed form 0 to 20 m depth. Using this assumption of Ekman drift and reanalysis winds, it is possible to reproduce the mooring velocity magnitude with 18% correlation. Whereas the 29% correlation between model and mooring data is unacceptable in most cases, it is still possible to argue that this is an improvement over the Ekman drift assumption. This is easily explained by the fact that the model captures more of the processes driving currents, including inertial oscillations.

In spite of the poor correlation between model and data, it captures many of the processes that drive net velocity, including Ekman, geostrophic, internal wave, inertial and subinertial processes. Consequently, we can use the model to determine the net drift and compare it to a typical geochemical assumption for Lagrangian drift. Here we refer to the approach employed by Rutgers Van Der Loeff et al. (2014). While evaluating radon deficits in the water column beneath variable ice cover, those authors assumed 0.5 cm/s bulk of mixed-layer speed and an elapsed time of 48 days to achieve a total drift of ~ 20 km.

The 20 km becomes the radius over which ice cover is averaged to account for variations in surface area for exchange. We computed the 48 day back trajectory of water parcels originating from Mooring D for two years using simulated velocities and found out that the average distance between the start and end points is 10.3 Km, this number is 49% less that what was mentioned on previous literature (Rutgers Van Der Loeff et al. 2014). This difference is due to water parcel trajectory deviating from a straight line and having a circular motion between the start and end point.

## 4. Summary

We have used a 36-km version of the MITgcm to evaluate whether coarse-resolution model output can be used to compensate for lack of data in the Arctic. The goal was to understand if/how to use model output to interpret geochemical tracer fields. This systematic comparison of upper ocean processes has revealed the following.

With regard to configuration, which was necessary to capture vertical profiles in near surface ocean, our experiments showed that changing the vertical resolution of model grid cells reduced the accuracy of the sea-ice trajectory and concentration. This is likely due to the fact that the model output was tuned with a different vertical grid spacing (Nguyen et al. 2011) and the tuning coefficients are no longer optimal, once we change that spacing. It has been shown by (Holloway et al. 2011) that optimized parametrization of the model is sensitive to horizontal resolution; vertical resolution may as well have the same effect on the model.

We observed good correlation (78%) between model ice velocity and ITP drift. The rapid accumulation of small errors in the drift speed and direction leads to significant differences in the predicted and actual drift trajectory. We found errors in drift trajectory of 2km/day. This relatively strong correlation is perhaps not surprising, considering that more than half of the ice



drift variability can be explained by Ekman drift alone, and the model receives wind as an input parameter. It is also worth noting that by doubling the model resolution, error decreases by more than half, i.e. 0.8 (km/day) accuracy by using a 18-km model (Nguyen et al. 2011).

The estimation of mixed layer depth is challenging: No algorithm performs well in all situations; CTD profiles from drifting buoys often do not include the top 7-10 m of the surface ocean where stratification can be important, and because the density structure of the ocean is affected by vertical fluxes and by geostrophy. In these model-data comparisons we found model MLD to be biased consistently shallower, but this in part depends on a surface value, which is not recorded in moored drifters such as ITPs. The evolution of the mixed layer showed that MLD correlates equally well (or poorly if constrained by sea-ice) with wind and temperature trends (36% and 39% respectively). Despite the potential for bias in the model MLD, the time series of ITP and model MLD reveal that the model captures variability at a similar frequency. It appears 10-50% changes in MLD happen regularly over the course of days and major changes (i.e. 100-200 %) appear to be event-driven as opposed to gradual seasonal evolution.

The potential for such event-driven changes in the MLD in the time prior to sampling are clearly important to consider when evaluating surface ocean geochemical tracer fields, although it may be sufficient to look for large excursions in the wind, air temperature and ice cover from reanalysis data.

The A1 experiment showed a good visual representation of Arctic general circulation but on finer scales, individual flows did not match the data from ITPs or Mooring. This discrepancy may be the result of two causes - the lack of eddy resolving resolution and the lack of realistic ice physics. Holloway et al. (2011) have shown that by increasing the resolution to less than 9 (km) a good agreement can be reached for 10 m vertical resolution. Our current lack of representation of velocity fields will lead to outcomes that are broadly similar, but locally incorrect. We further saw this effect on temperature profiles modeled on ITP path.

We compare each of the variables affecting gas-exchange and summarize our comparison on Table-1. Overall, we find that a coarse-resolution model can yield as much as 50% improvement over simple assumptions that are loosely constrained by data (e.g. assuming Ekman drift or using a radius to average ice properties over a given time period). However, the coarse resolution model is not up to the task of representing Lagrangian drift of a water parcel. Water parcel drift beneath sea ice remains an elusive term, difficult to estimate, and hard to do without, when interpreting geochemical tracer fields.



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



# Figures

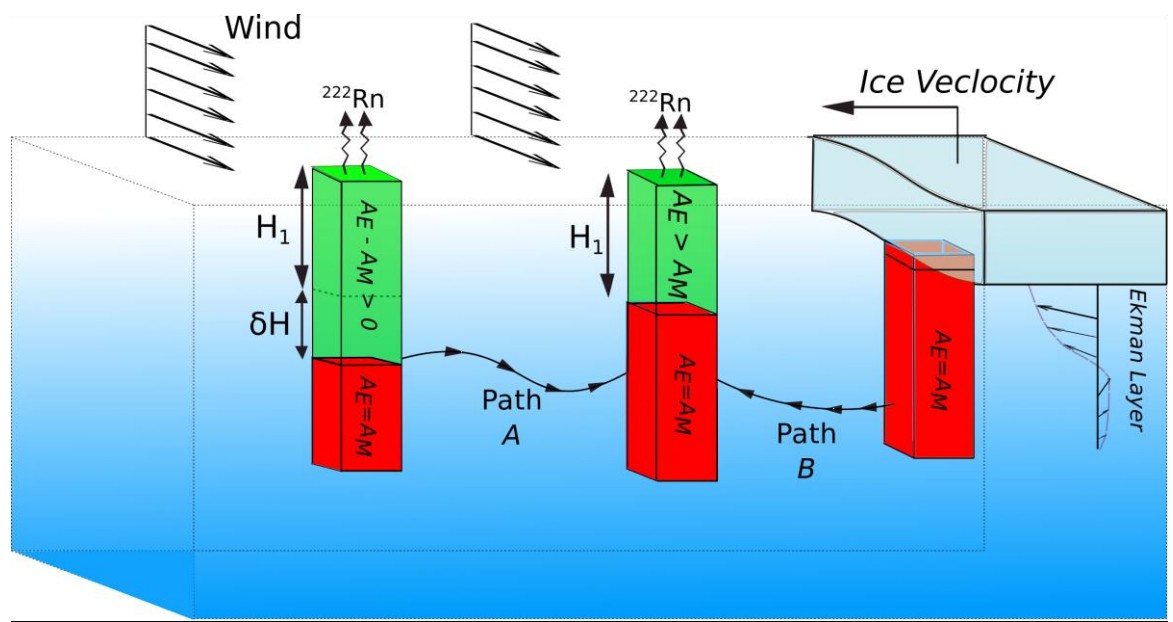

Fig 1. A graphic illustration of two possible back trajectory of a single sampling station

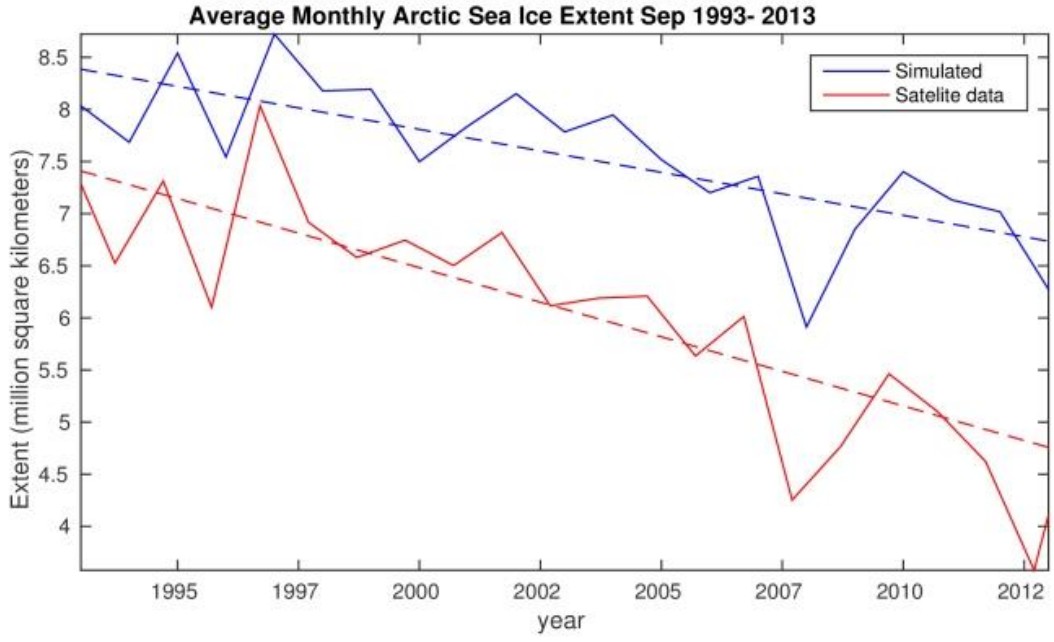

Fig 2. Arctic Sea-ice extent from 1992-2012 using A0





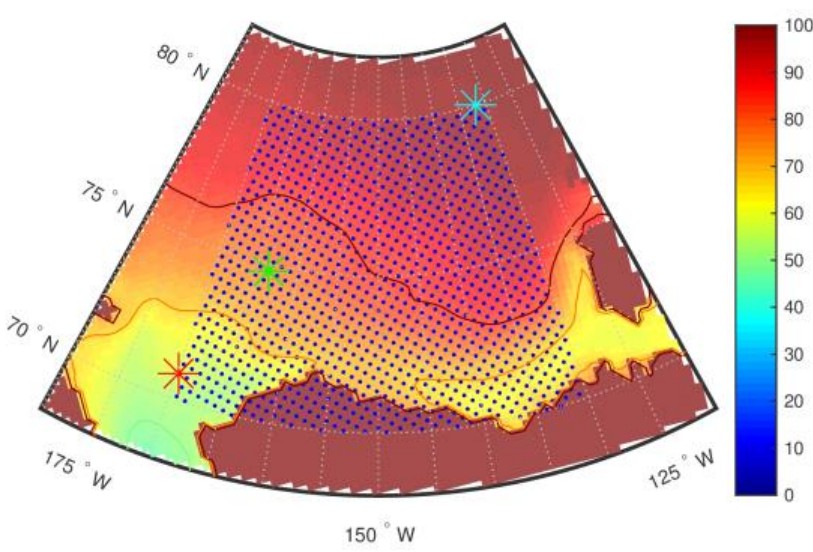

Fig 3. Averaged satellite sea ice cover from 2006-2013, Solid red line marking 80% cover and orange line marking 60%, Blue dots show the analysis grid, stars show the location of the three points Cyan P1, Green P2, Red P3 where time series data is graphed in Figure 3.

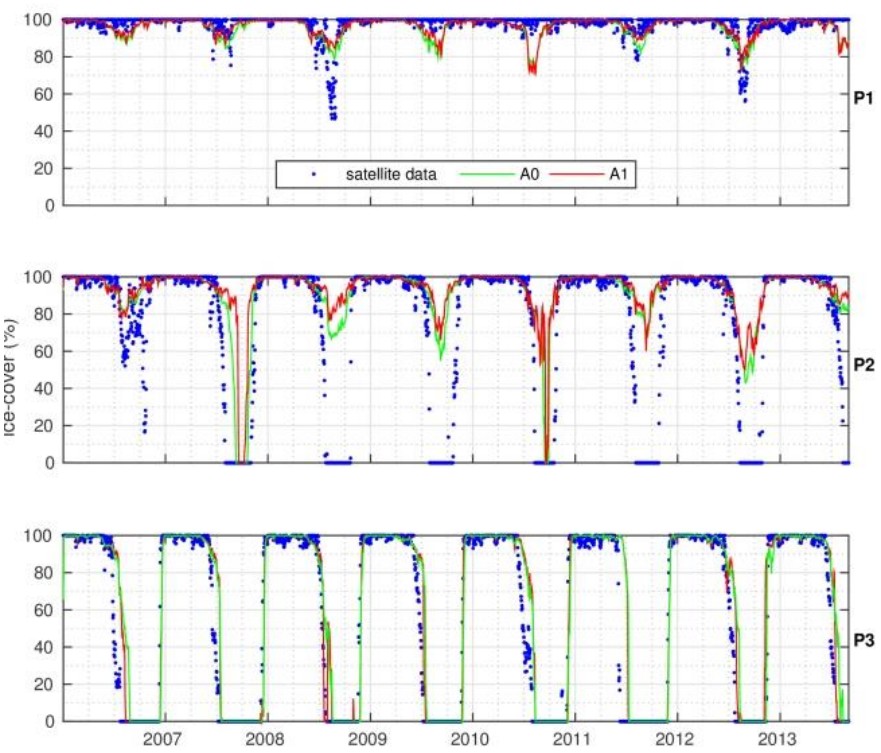

Fig 4. Time history of Sea-Ice fraction from top P1, P2 and P3, Satellite data compared with model results A0 and A1





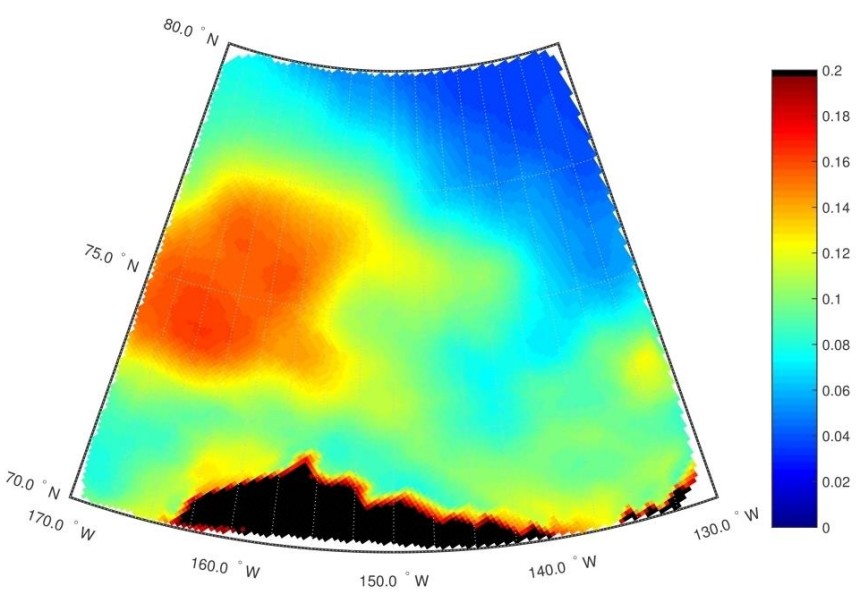

Fig 5. Root mean square error averaged of A1 over time from 2006 to 2013; black mask covers the grid points on the ground

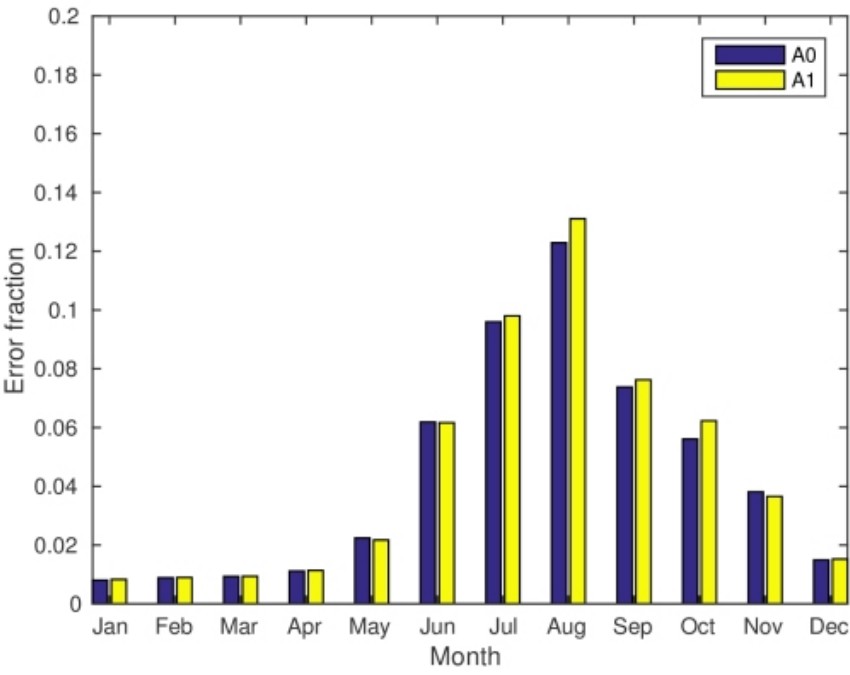

Fig 6. Monthly averaged Annual Root Mean Square Error



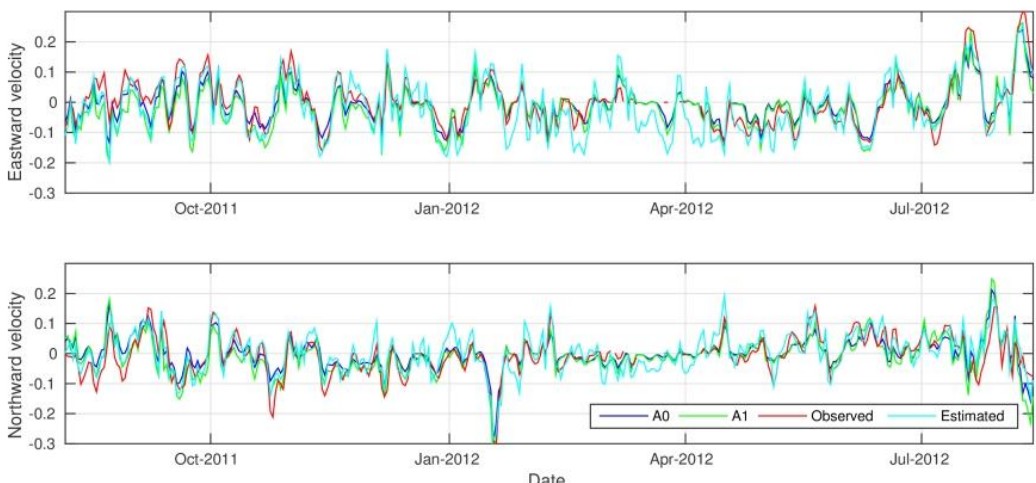

Fig 7. Velocity components from A0 and A1 compared with drift velocity of ITP-53

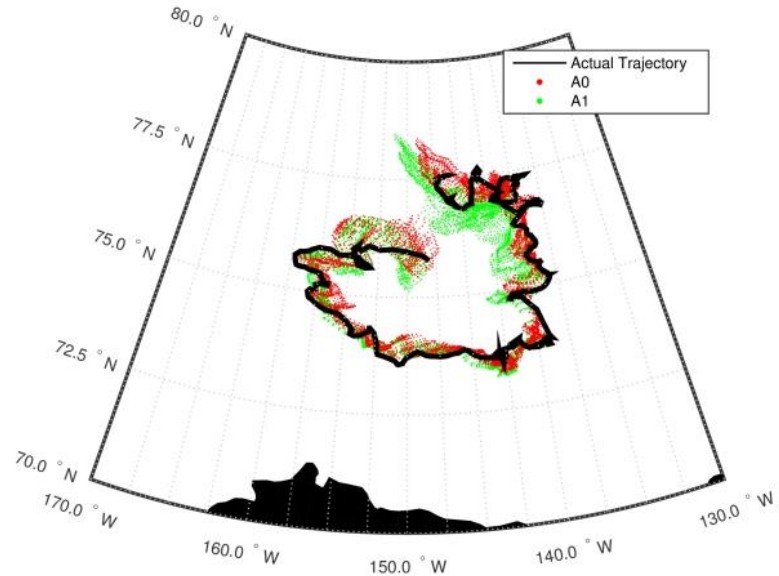

5                                Fig 8. 30 day trajectory of ITP 53, starting August 2011 to August 2012



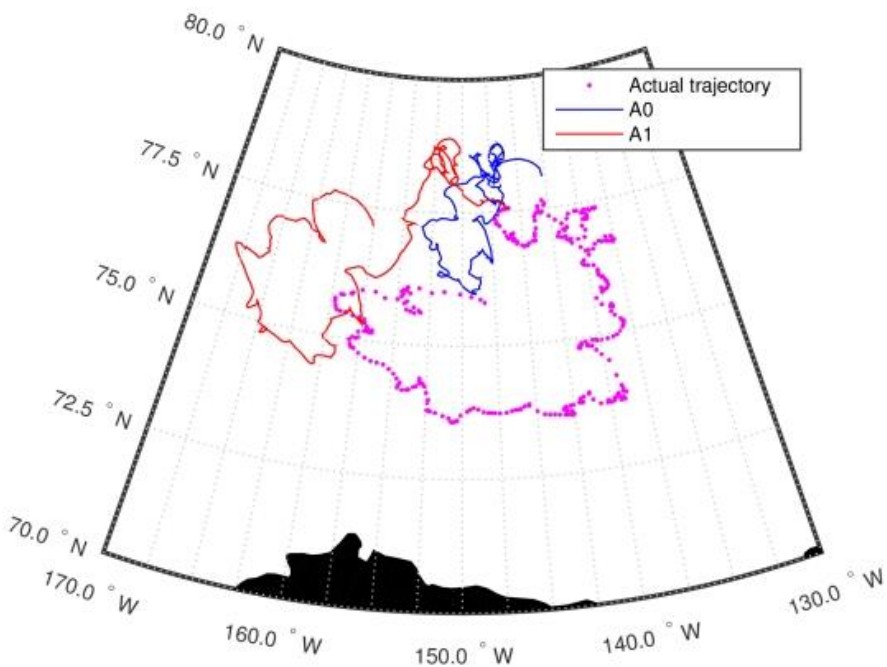

Fig 9. Simulated and observed trajectory of ITP53, Aug 06 2011 to Aug 13 2012

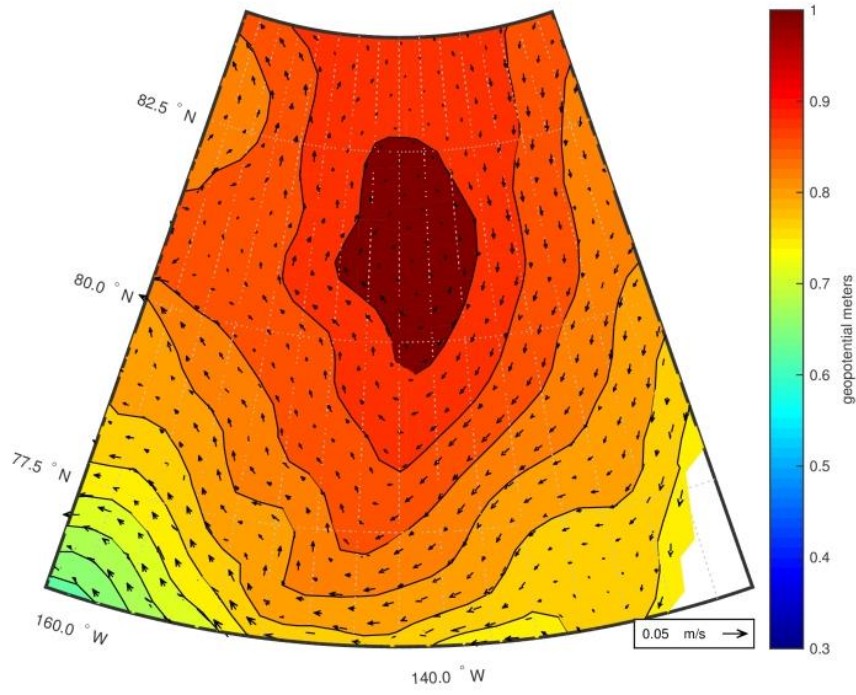

Fig 10. MITgcm geopotential height referenced to 400(m), 2008-2011





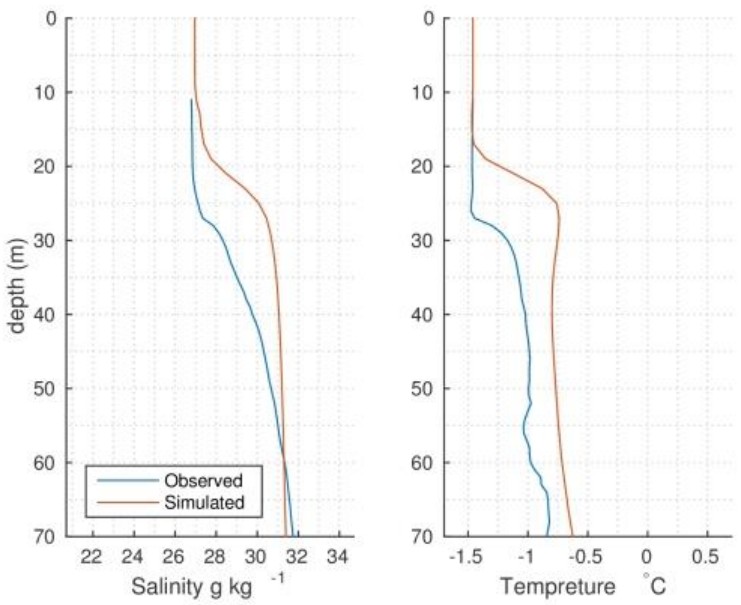

Fig 11. Obtained from ITP 1 on 13-Dec-2006 at 74.80°N and 131.44°W

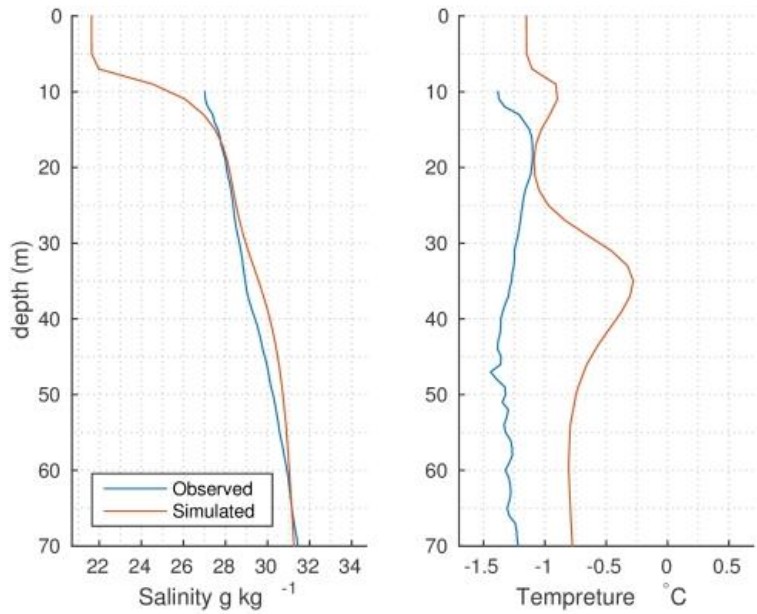

5                          Fig 12: Obtained from ITP 1 on 28-Aug-2006 at 76.96°N and 133.32°W




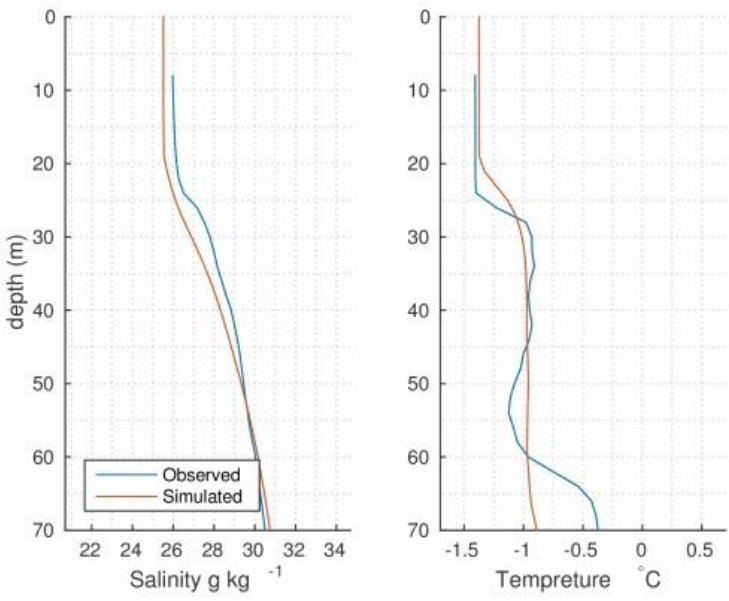

Fig 13. Obtained from ITP-43 on 27-Nov-2010 at 75.41°N and 143.09°W

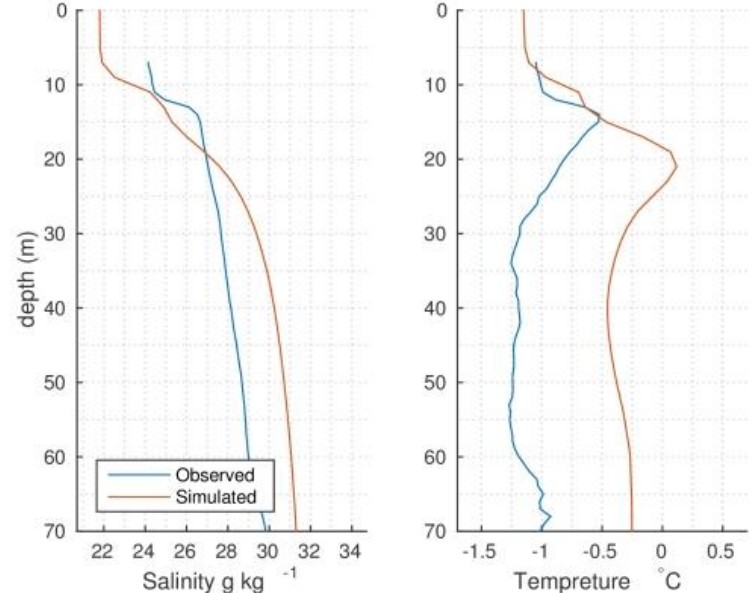

Fig 14: Obtained from ITP-13 on 30-Jul-2008 at 75.00°N and 132.78°W



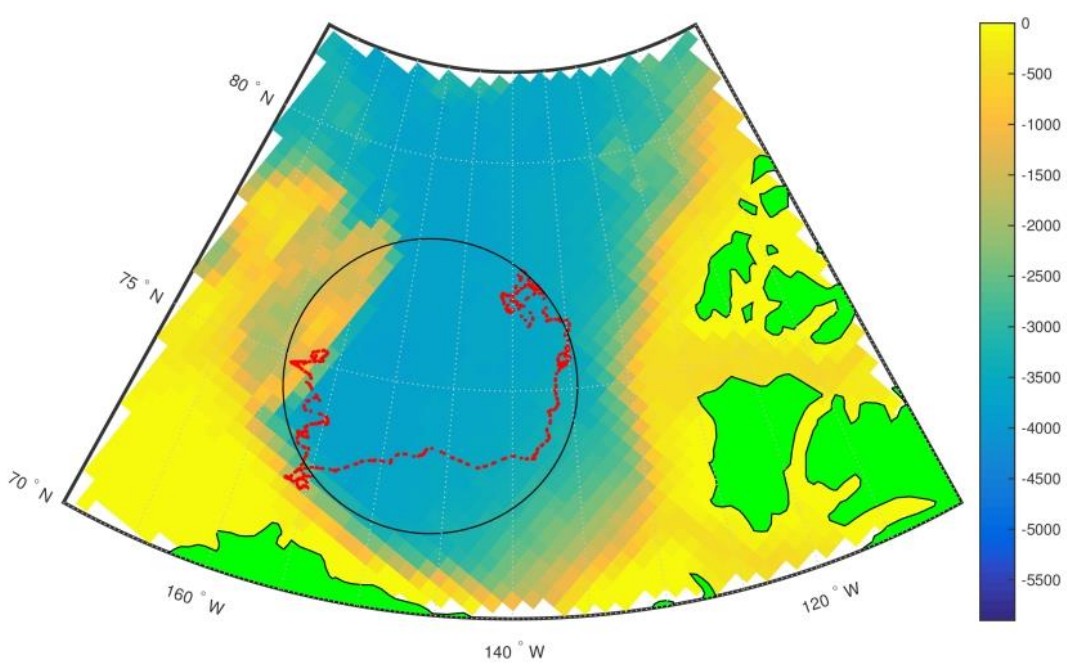

Fig 15. ITP-62 distribution of sampling stations and the circle centered at 75N 147 W and radius of 330 (km)

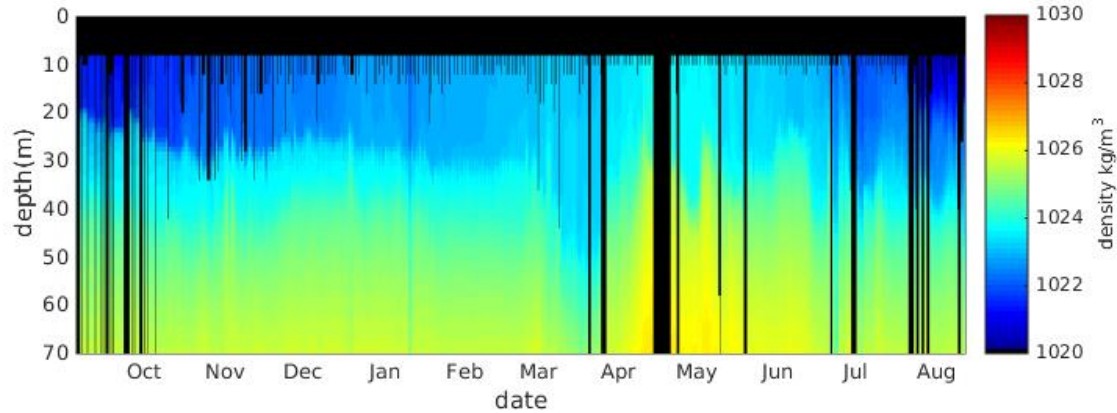

Fig 16. Observed upper ocean density, ITP 62 Sep 04 2012 to 12 Aug 2013



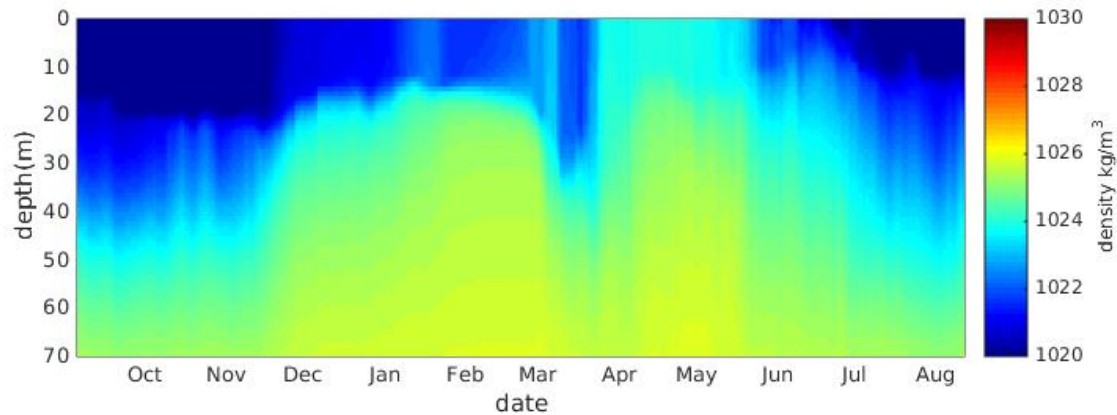

Fig 17. Simulated upper ocean density following ITP-62

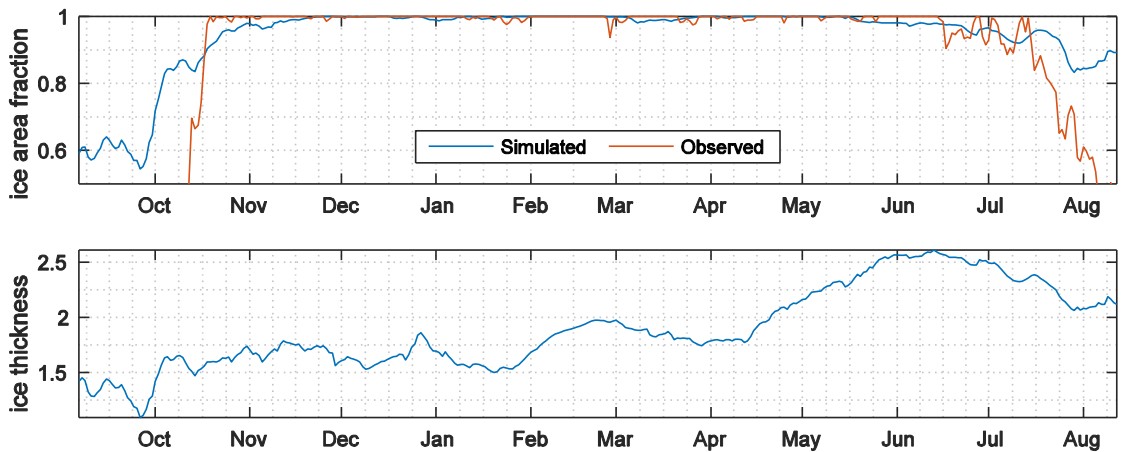

Fig. 18: Sea Ice fraction and Modeled Ice thickness interpolated into ITP-62 drift



Fig 19: Different methods applied to simulated density profiles from a)ITP 1 winter b)ITP 43 winter c)ITP 1 summer d) ITP13 summer



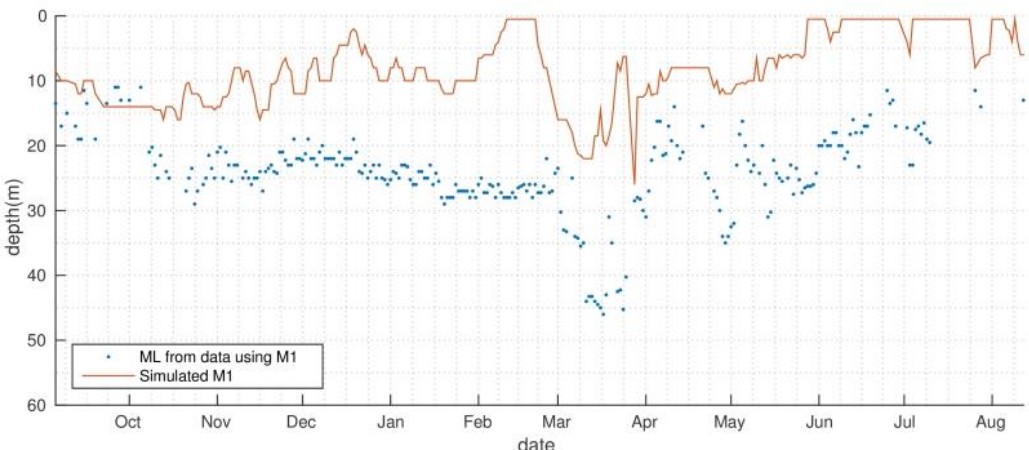

Fig 20: Mixed layer depth time series based on M1, simulation and data from ITP-62

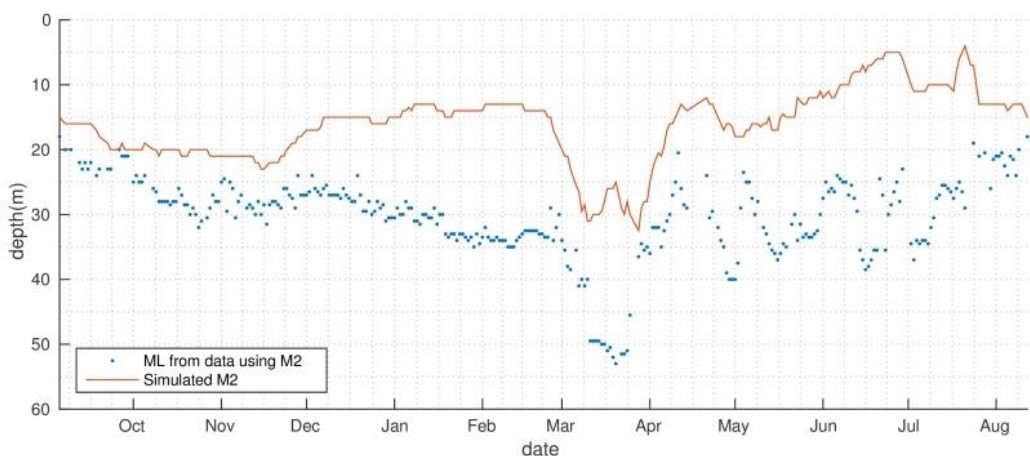

Fig 21: Mixed layer depth time series based on M2, simulation and data from ITP-62





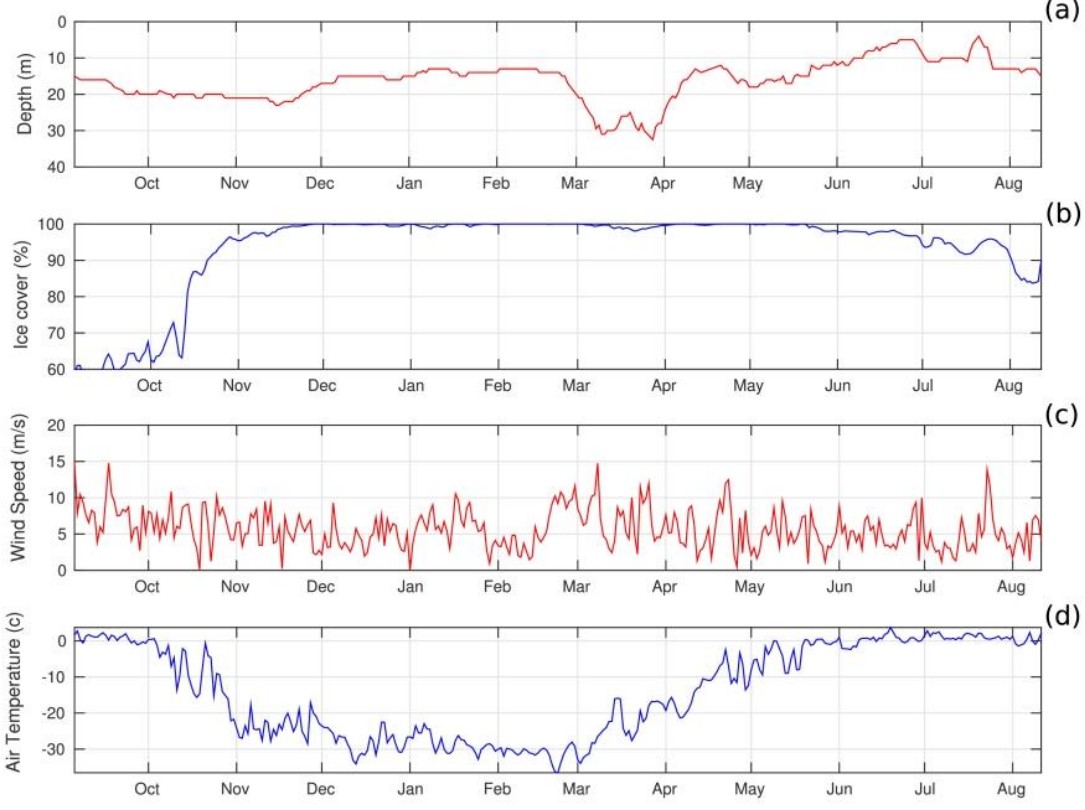

Fig 22: a) Mixed layer evolution during ITP 62 drift b) Sea-Ice fraction c) Wind Speed 10m above the surface d) Air temperature





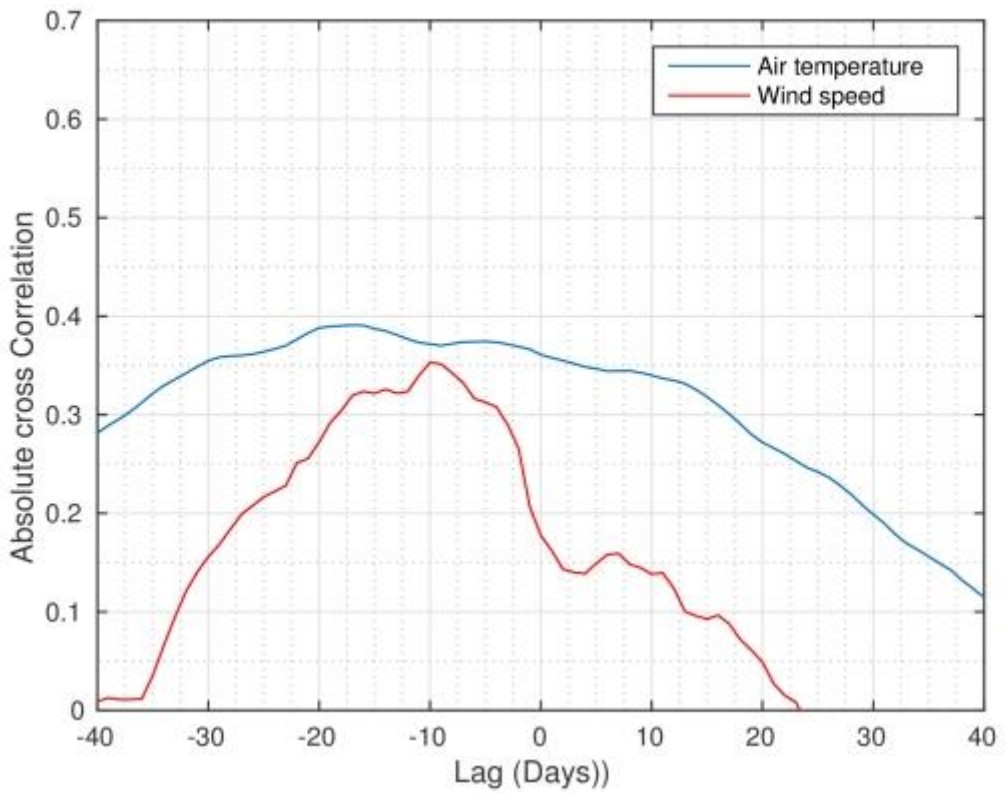

Fig 23: Cross Correlation of Mixed layer depth with wind speed and air temperature





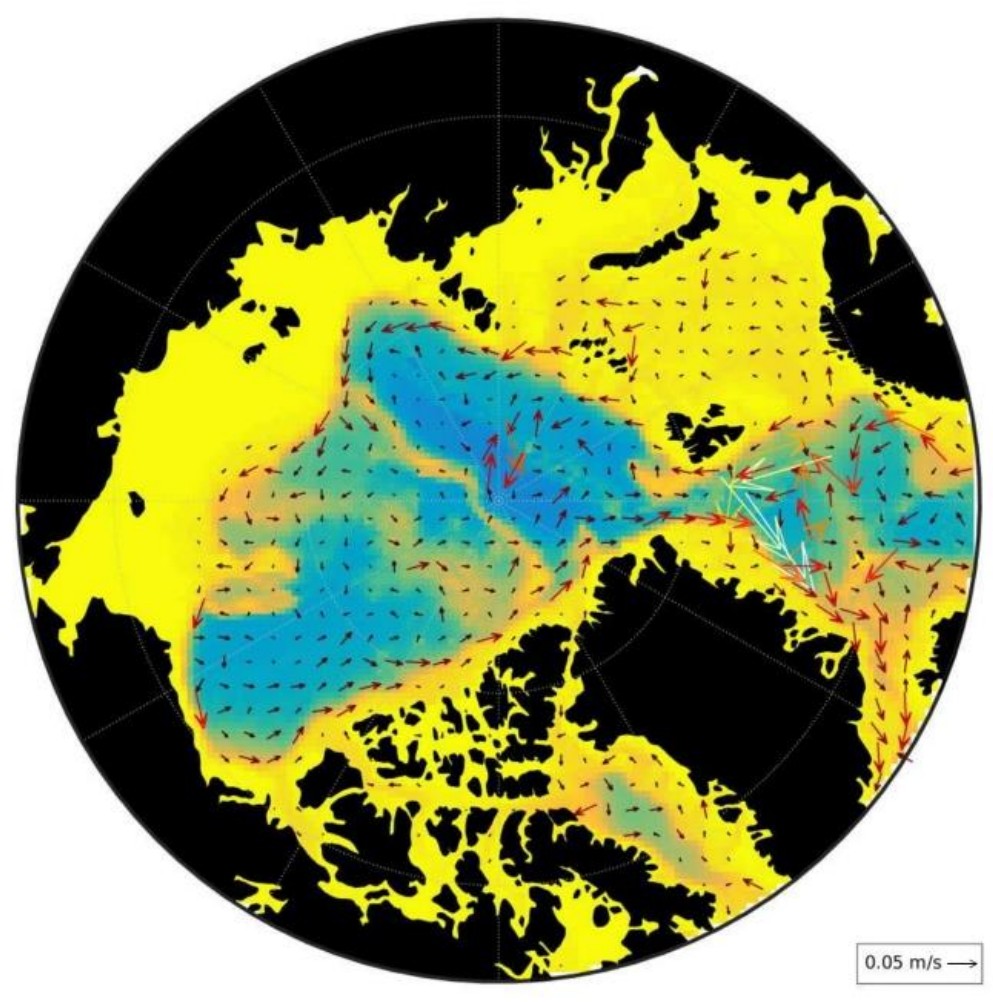

Fig 24. Model velocities averaged at 180-250(m) , 2006-2012





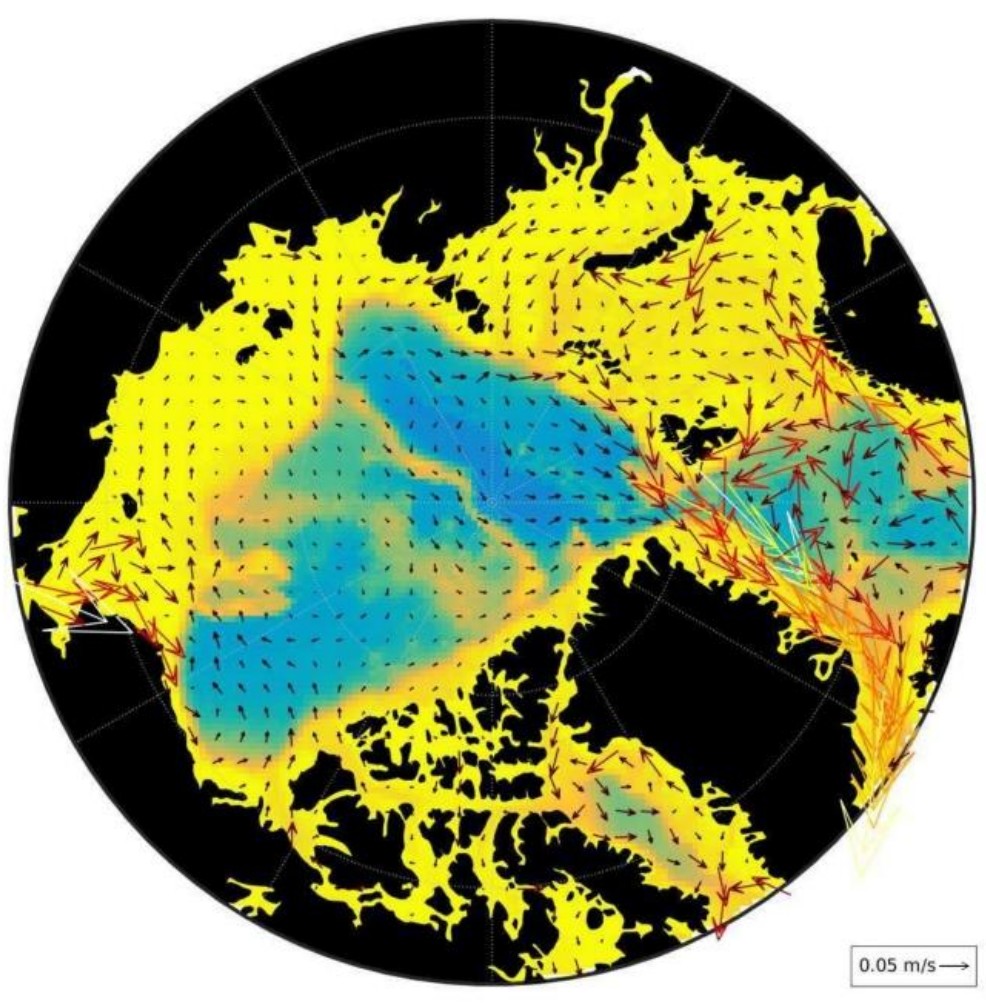

Fig 25. Model velocities averaged from 1-15 (m), 2006-2012



Fig 26. ITP-V Speed simulated and observed

Fig 27. ITP-V Eastward velocity observed and simulated





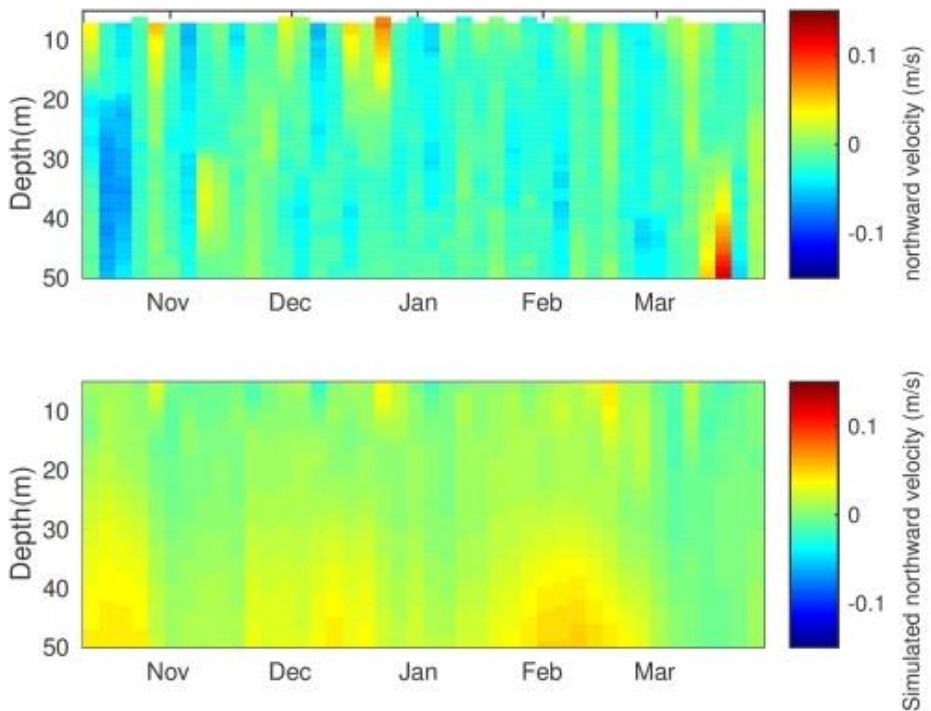

Fig 28. ITP-V Northward velocity observed and simulated

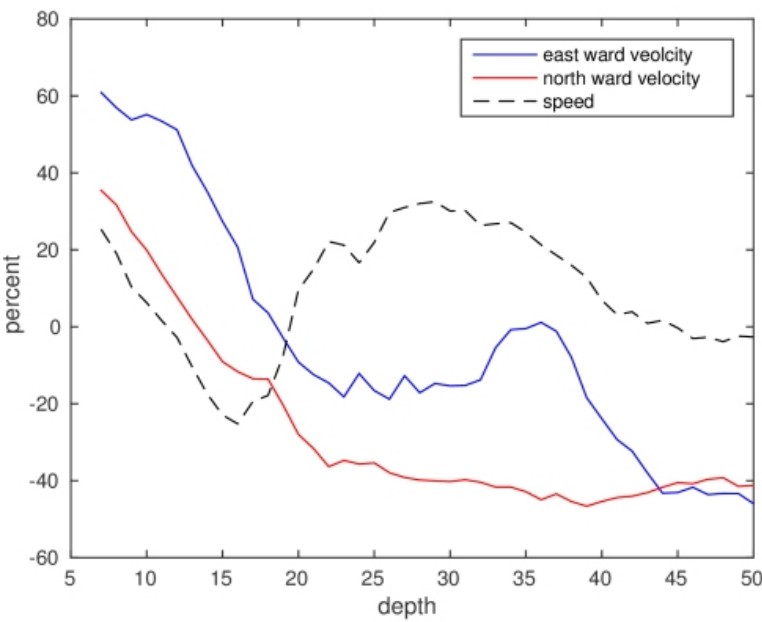

Fig 29. ITP-V correlation versus depth





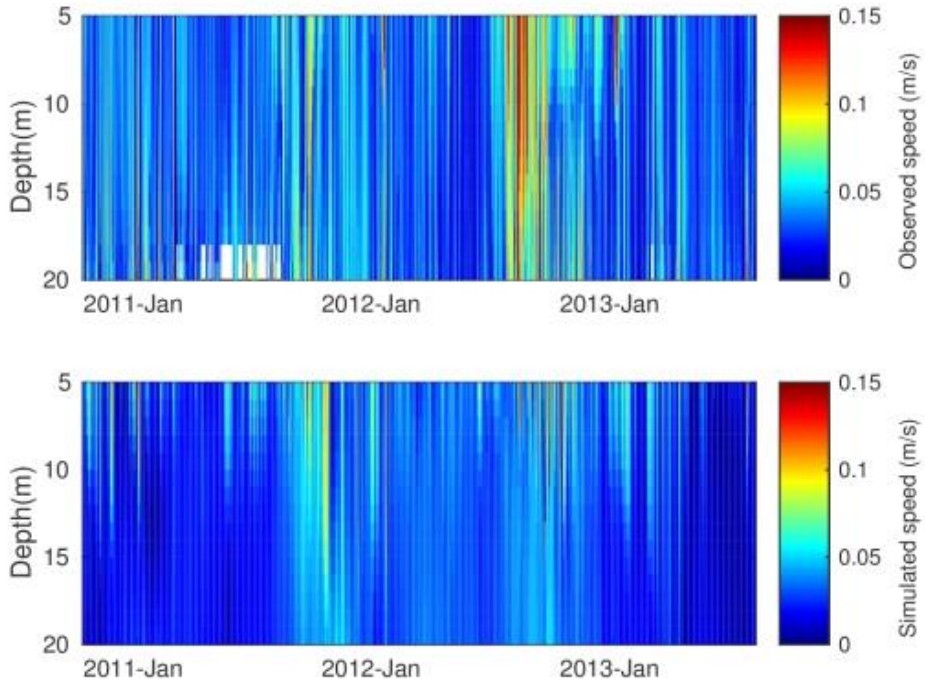

Fig 30. Mooring-D Observed and simulated Speed





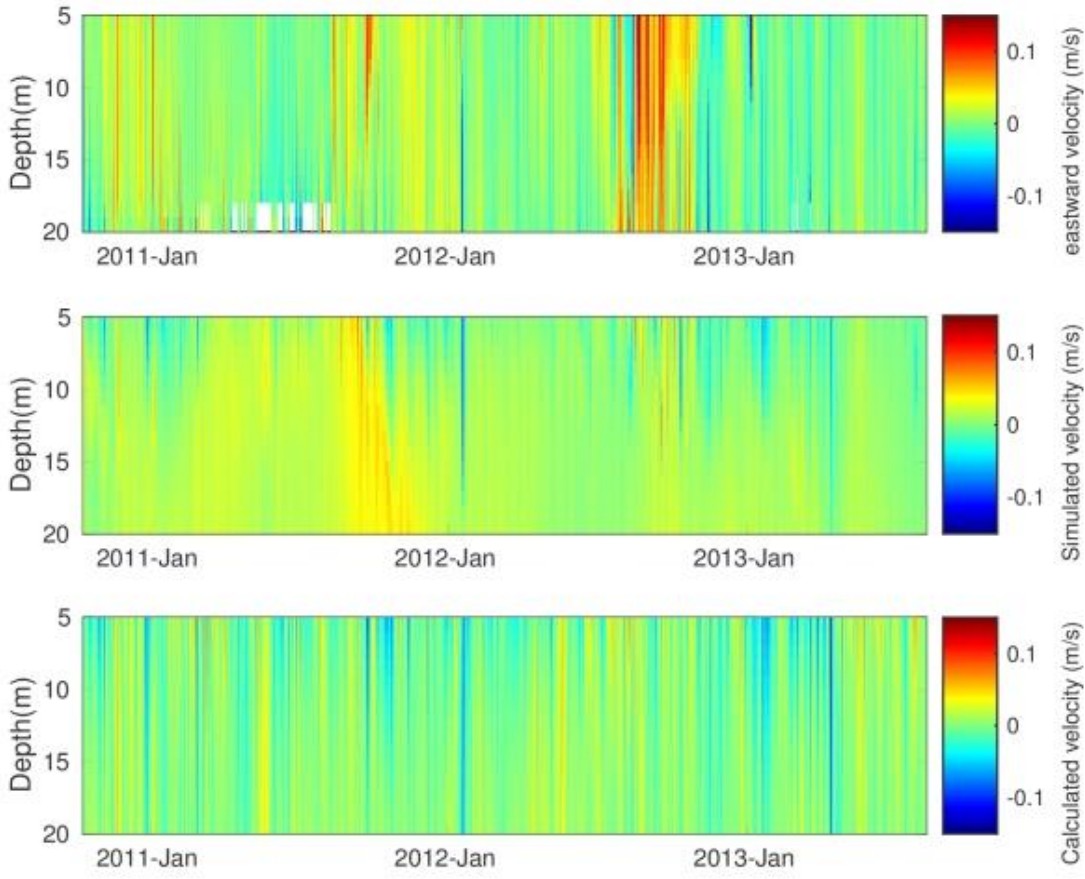

Fig 31. Mooring-D eastward velocity observed, simulated and estimated





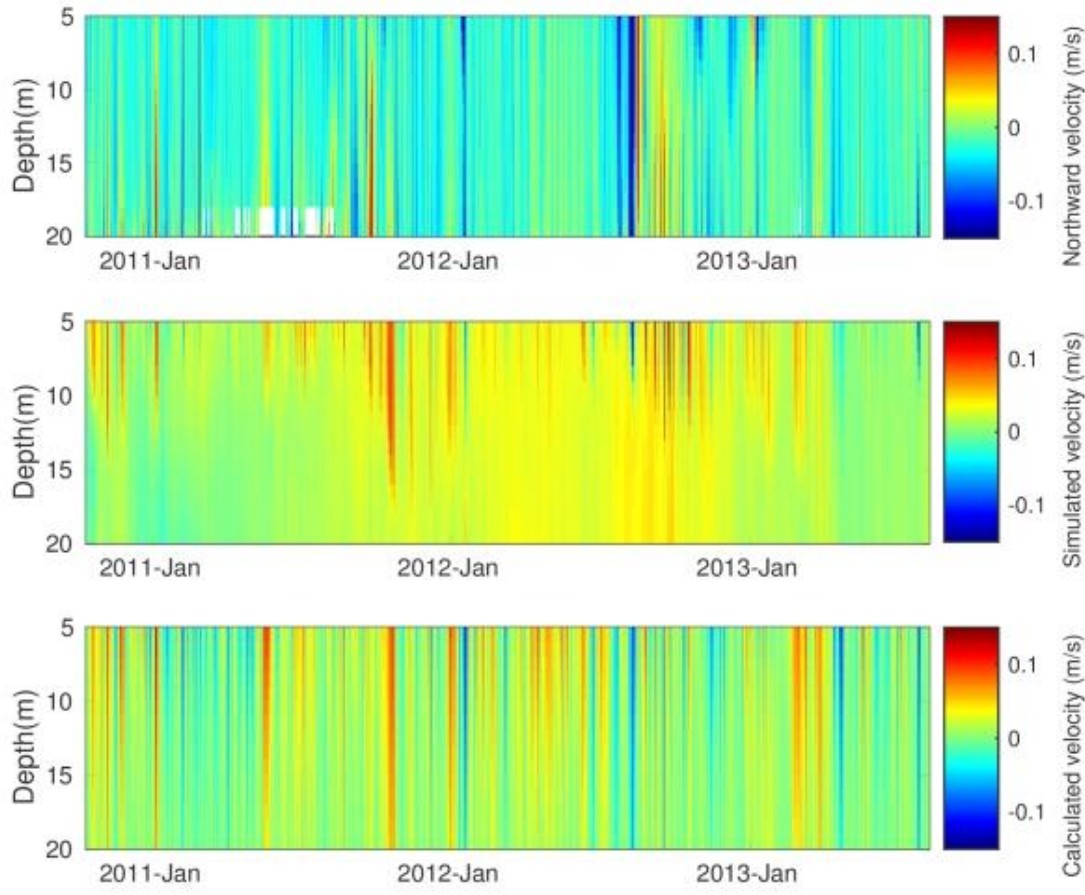

Fig 32. Mooring-D northward velocity observed, simulated and Estimated





**Tables**

|  | **Sea ice fraction** | **Sea ice drift** | **Sea ice velocity** | **Mixed layer depth** | **Water current velocity for back trajectory calculation** |
|---|---|---|---|---|---|
| **Coarse resolution 3D simulation** | Between 98% to 88% correlation | 2 Km d$^{-1}$ error | 78% correlation | Model follows mixed layer evolution trend | 29% correlation between model velocity and mooring data |
| **Estimation from direct observation** | Satellite data readily available | Remote sensing estimates available. Coverage is spotty. | Remote sensing estimates available. Coverage is spotty. | Mooring or ITP, but not available inside Lagrangian water parcel | Mooring or ITP, but not available inside lagrangian water parcel |
| **Estimation from simple assumptions** | No need for estimation | 4.2 Km d$^{-1}$ error | 65% correlation | Not quantified | 18% correlation between Ekman flow and mooring data. |
| **Take home message** | Use satellite data | Use satellite data where available. Model trajectories may have value as ensemble averages | Sea ice velocity from model has a good correlation and is valuable | Absolute value of MLD can be biased, but model can reveal the potential for changes | Ekman assumption is easier to implement, albeit less accurate by 11%. Simulations indicate a radius (10.3 km) for averaging properties. |

Table-1 Summary of comparison between the simulation, estimation and data