# Peer review of "Numerical investigation of the Arctic ice-ocean boundary layer; implications for air-sea gas fluxes"

_Ocean Science, 2016_

## Referee Comment (RC1) · Anonymous Referee #1 · 7 Sep 2016

This paper is interested in the ice ocean boundary layer. The long term goal is to focus on air-sea gas fluxes. These air-sea fluxes depend on water column properties, and their history. Given that such data is often lacking, the authors examine if they can use output from a coarse resolution numerical models. Air-sea fluxes are not estimated, but instead the authors compare model fields of sea ice, upper ocean properties and mixed layer depth with observations, such as from ice tethered profilers. The analysis shows the ice concentration and velocities from the model compare well with the observations. For the mixed layer depth, broad trends are comparable. But significant biases in upper ocean properties and velocities are found.

The underlying issue for this paper, determining if model fields can be used in the

estimation of air-sea gas fluxes is interesting and important. However, I don't believe the present paper does a good job of answering the question posed. Rather than taking the output from the latest and most advanced ocean general circulation models, a number of which have been extensively evaluated in the Arctic, the authors try to set up and run their own coarse resolution simulations. Beyond some specific issues about the author's experiments and analysis, I don't see the purpose of these stand-alone experiments. Given the groups doing high resolution ocean modelling have the spent the time improving and evaluating their models, I don't see why such fields aren't being obtained and considered for this analysis. Additionally, since any given model has issues, doing this analysis with an ensemble of models would give much better confidence for the observational comparisons and thus the utility of numerical model output in calculating air-sea gas fluxes.

Additionally, this is sloppily prepared manuscript. Almost all the figures are mis-numbered in the text. Most of the figure captions are too brief and lack the detail needed to clearly explain all aspects presented. The details on the figures are also not well explained in the text. A number of figures have spelling errors on the axes labels. 32 figures are probably too many as well. Also, there are some technical issues with respect to the use and analysis of the numerical model output.

Thus I can't recommend that this paper be accepted in its present form. I would suggest that the authors get high resolution model output from colleagues (ideally more than one model) and then repeat their analysis with that output. Then I think the authors may have a really interesting paper, that will be really useful for the community.

I won't bother with text comments given that text will significantly change if the paper is revised. Instead I will provide a few additional specific comments below that the authors may find useful in thinking about the topic and revising their manuscript.

Introduction: With the later focus on the Beaufort Gyre, it needs to be discussed (especially the observations that will later be used) in the introduction. The reader has

otherwise no idea of the geographical focus until much later in the paper.

Page 3, Line 1: In areas of deep convection, the mixed layer can change by much more than a factor of 2 over several weeks.

Page 3, Line 17: Why do you need to use output from a model run on a desktop? Being able to run a model is a very different task compared to running a model to produce high quality evaluated output. Just take the fields from those produced by modelling groups.

Page 4, line 14: Era40 is the output from an atmospheric reanalysis. This is not the same as a numerical weather forecast.

Page 4, Line 19: Most high resolution models now use vertical resolution of 1-3 m near the surface.

Page 10, Line 8, etc.: Do not understand how the MLD could be zero. Model tracer points are in the middle of the grid cell for the MIT model. Thus, I don't see how the MLD could less than the depth of the first layer, given M1 is based on a threshold approach.

Page 10, line 11: Might it not be useful to use a simple 1-D mixed layer model to examine directly the impact of the forcing.

Figures 26-32: I'm surprised the model and ITP velocities differ by so much. Have the model velocities been rotated from the model grid to latitude-longitude?

---

## Referee Comment (RC2) · Anonymous Referee #2 · 2 Nov 2016

The aim of the paper is to examine if the mixed layer depth from the regional ocean model configurations of Arctic is accurate enough for the estimation of the air-sea gas exchange rates. However, the authors spend majority of paper evaluating the sea-ice velocity, sea-ice concentration and temperature and salinity profiles against observations from satellite, moorings and ice-tethered profilers. I found that the motivation and some of contents in the paper are not aligned. I would not accept the paper for publication in its present form. The paper needs to be re-organized to address the following issues:

1.  The motivation of the paper is to estimate the mixed layer depth from the ocean model; however, the following sections do not seem to have any connections to the

mixed layer depth estimation: 3.3.1 Geopotential height, 3.31 Vertical salinity temperature profiles and 3.6 Circulation and etc. This makes the interpretation of this paper difficult and results in many figures, which are not labeled correctly. For example, line 20 describes the sea-ice trajectory, but Fig. 10 contains color plot of geopotential and arrows, which are not described. Line 25 describes the geopotential, yet the referred figure 12 does not contain any information on the geopotential. I would keep the section 3.5 Mixed layer depth and omit 3.3.1, 3.6 and 3.7 unless there are clear connections to the change in the mixed layer depth.

2. The paper claims in the abstract "Overall, we find the course resolution model to be an inadequate surrogate for sparse data, however the simulation results are a slight improvement over several of the simplifying assumptions that are often made when surface ocean geochemistry, including the use of a constant mixed layer depth and a velocity profile that is purely wind-driven." I agree with the first claim "find the course resolution model to be an inadequate surrogate for sparse data"; however I do not see where the second claim, "the simulation results are a slight improvement over several of the simplifying assumptions that are often made when surface ocean geochemistry", is supported in the paper. The authors need to show the rate of air-sea gas exchange calculated from the mixed layer depth from 1) the ocean model results and 2) the conventional method, and discuss the difference in the estimates.

3. I am concerned with the use of salt plume parameterization in the model configuration, A1. The A1 has relatively high vertical resolution of 2 m down to upper 50 m, whereas the salt plume parameterization is tested for models having the vertical resolution of ~10 m. The salt plume parameterization is known to suppress the mixing to maintain a reasonable mixed layer depth in the polar regions. The parameterization is meant for a coarse vertical resolution model (~10 m). The vertical resolution in A1 is 5 times higher, and I am concerned that the parameterization in A1 is over suppressing the mixed layer depth. In fact, Figs 11, 13 20, 21 all points to the underestimation of the mixed layer depth compared to the observations. I suggest switching off the salt plume

parameterization and evaluating the mixed layer depth in comparison to the existing model results and observations.

Minor comments

Line 10-15: It is unclear if the lateral boundary conditions are prescribed from the climatology or seasonal output from ECCO2 and JRA25. It is also unclear why the authors use output from two different models ECCO2 and JRA25. Map of the domain showing bathymetry will be helpful.

Equation 3: What is the definition of i?
* * *

---

## Author Comment (AC1) · 3 Dec 2016

We appreciate the thoughtful input from both of the reviewers, and we feel that we did address all the comments and suggestions that the reviewers have made on MS NO.:os-2016-4. Below, our point by point response is formatted as reviewer suggestion (R), author response (AR) and changes to manuscript (MC):

Reviewer #1

R1-1)"The underlying issue for this paper, determining if model fields can be used in the estimation of air-sea gas fluxes is interesting and important. However, I don't believe the present paper does a good job of answering the question posed. Rather

than taking the output from the latest and most advanced ocean general circulation models, a number of which have been extensively evaluated in the Arctic, the authors try to set up and run their own coarse resolution simulations. Beyond some specific issues about the author's experiments and analysis, I don't see the purpose of these standalone experiments. Given the groups doing high resolution ocean modelling have the spent the time improving and evaluating their models, I don't see why such fields aren't being obtained and considered for this analysis. Additionally, since any given model has issues, doing this analysis with an ensemble of models would give much better confidence for the observational comparisons and thus the utility of numerical model output in calculating air-sea gas fluxes. "

AR-MC) The reviewer makes a valid point about model resolution. Initially, our goal was to use a model that could be run on a desktop, but since that time the resolution question has been recurring. Therefore, we have utilized other model resolutions, including model output from An. T. Nguyen (now a co-author and expert on MITgcm) and included those results in this revision. Hence on sections that we felt that 36km model does a poor job of following the data, we added 9km and 2km horizontal resolution. It is apparent from these new model runs that increasing the resolution did not show improvements in its capacity to match the field data from moorings and ITP. We believe this discrepancy is due to input forcing of the model (ie reanalysis data products); specifically the low correlation in wind and air temperature reanalysis products on high frequencies which would make the accuracy of model invariant in respect to resolution.

R1-1)"Introduction: With the later focus on the Beaufort Gyre, it needs to be discussed (especially the observations that will later be used) in the introduction. The reader has otherwise no idea of the geographical focus until much later in the paper."

AR) We agree with R1 that the geographic boundaries of our comparison should be mentioned and we addressed it.

MC) added Figure 2. Page 5 Paragraph 3: "Most of the observed data exist in Beaufort

Gyre, hence we mostly focus our comparison to that geographic perimeter. Figure 2 depicts the bathymetry and location of most important observations we used to make the comparisons with the model."

R1-2)"Page 3 Line 1: In areas of deep convection, the mixed layer can change by much more than a factor of 2 over several weeks."

AR) For the purpose of back tracing Radon-Labeled and gas exchange forcing acting upon it, time scale of interest is ~15-20 days, nevertheless we agree that it should be mentioned.

MC) Page 8 , line 4 "Between February and March, ITP-35 appears to drift through a zone of convection zone, likely caused by ice formation, with sudden increase of density near the surface. The same feature can be observed in both A1 and A2 density. However, on a smaller scale, there is significantly more variation in the ITP data than what the model represents."

R1-3)" Page 3, Line 17: Why do you need to use output from a model run on a desktop? Being able to run a model is a very different task compared to running a model to produce high quality evaluated output. Just take the fields from those produced by modelling groups."

AR) As stated above, we added the 9km and 2km models which are extremely computational heavy models to our manuscript. Specifically, we added 9km to mixed layer depth section and 9km and 2km models to water velocity section and removed "run on desktop".

MC)Page 4, Line 9 "Three ECCO's configurations are used, at horizontal grid spacings of 36 km, 9 km, and 2 km, respectively. The models are based on the Massachusetts Institute of Technology general circulation model (MITgcm) code and employ the z coordinate system described in Adcroft and Campin (2004). Our approach is first to assess the model outputs from the coarse resolution model using model-data misfits,

then to investigate if there are quantitative reductions in model-data misfits with higher horizontal resolutions."

R1-3)"Page 4, line 14: Era40 is the output from an atmospheric reanalysis. This is not the same as a numerical weather forecast."

AR) The reviewer makes a very good point that we are not addressing the atmosphere products correctly, we fixed that.

MC) Page 4, line 13 "Surface forcings are from the 25 year Japanese Reanalysis Project (JRA25)(Onogi et al. 2007) for 36 km and 9 km runs and The European Center for Medium-Range Weather Forecasts (ECMWF) analysis for 2 km run"

R1-4)"Page 4, Line 19: Most high resolution models now use vertical resolution of 1-3 m near the surface."

AR) We removed the coarse resolution model with 10m vertical spacing from the figures and discussion and stated that our 36km and 9km share a 2m near surface vertical resolution and 2km model start with 1m vertical resolution near surface.

MC)Page 4 Line 23 "We introduced a set of new vertical grid spacings to allow us to capture near surface small details which cannot be represented with the coarser grid system. In the 36 km (hereafter referred to as A1) and 9 km (called A2) models, the spacing is 2 m in the upper 50 m of the water column and gradually increases to a maximum of 650 meters. In contrast, the 2 km model (called A3) has 25 layers in the top 100 meters of water column, starting from 1 meter and increasing to 15 meters step. "

R1-5) "Page 10, Line 8, etc.: Do not understand how the MLD could be zero. Model tracer points are in the middle of the grid cell for the MIT model. Thus, I don't see how the MLD could less than the depth of the first layer, given M1 is based on a threshold approach."

AR-MC) The zero mixed layer depth has been a byproduct of extrapolation of results

from last grid point i.e. 1 meter below the surface to the surface. We agree with reviewer that such extrapolation is not necessary and we confined our analysis to first grid and removed any extrapolation. We also, in favor of limiting the number of figures, refrained to repeat our analysis for two methods of mixed layer calculation for ITP profiles and chose the M2 method for ITP mixed layer analysis. Aiming for consistency, we also changed the focused ITP from ITP-62 to ITP-V(35) for mixed layer depth discussion.

R1-6)" Page 10, line 11: Might it not be useful to use a simple 1-D mixed layer model to examine directly the impact of the forcing."

AR)One definite advantage of coarse resolution model over the 1D model is the ability of coarse resolution model to produce sea ice advection. On 1D model the amount of sea ice on top of the water column is a function of heat budget. Ice that is formed on far field and moved by wind to top of the water column can affect the mixed layer and is neglected in 1D model. To stress the importance of ice advection and ice openings by wind divergence and its effects on mixed layer change, we added a figure with 9km and 36km sea ice outputs on the path of ITP to mixed layer section and discussed the results.

MC) added Figure 8, Page 9 Line 14 " The difference between A1 and A2 and their ability to capture MLD change, can be explained by the capability of a higher resolution model to capture small-scale fractures in the ice cover (Figure 8), and conversely, the inability of the coarser resolution to do so is due to averaging over a larger grid. The wind appears to be the primary driving mechanism for the divergence in ice cover, which in turn exposes the ocean to the cold atmosphere and lead to a loss of buoyancy and an increase in MLD. With higher resolution these openings can be captured, leading to a better agreement with data in marginal ice zones"

R1-7) Figures 26-32: I'm surprised the model and ITP velocities differ by so much. Have the model velocities been rotated from the model grid to latitude-longitude?

AR) We understand and shared the reviewers concerns over the accuracy of water

velocity outputs. We checked and made sure that the velocities have been rotated and our rotation mechanism is sound, as seen by our sea ice velocity outputs. By comparing with higher resolution i.e. 2km and 9km, we made sure no improvement can be reached by increasing resolution either. We suspected and investigated the wind forcing. Chaudhuri et al. (2014) has published a comparison of all reanalysis products with data in Arctic and observed that none of the wind products has a correlation of more than 0.2. This low correlation propagates into water velocities. We added 9km and 2km outputs to water velocity section and stated the effects of reanalysis wind products on near surface velocities. Also in favor of coherence, we limited our comparison to a single level for mooring data and a vertical average velocity between 5 to 50 meters for ITP-V data. We also removed the subsection on Ekman analysis based on wind from water velocity section. Since we suspected the error to be originating from wind inputs, calculation of Ekman layer based on same wind was unnecessary.

MC)Figure 9 added ; Page 10 line 9 "We further add A3 to our comparison for moorings velocities (Figure 9), and compared velocities at 25m, which is the level that is shared between all our models and removes the necessity of any interpolation. The simulation results show RMSE normalized by data of higher than 5 and correlations of less than 0.3 over 3 moorings and almost two years of data. This result indicates ocean currents are not well captured in the model irrespective of horizontal grid resolution. We must therefore look into the atmospheric forcing as a likely source of error on high frequency water velocities near surface. As noted above, the wind inputs into the model from the reanalyses are available at frequency 6-hourly. (Chaudhuri et al. 2014; Lindsay et al. 2014) have compared various available reanalysis products over the Arctic which we used to force our model, along with multiple other reanalysis products with available ship-based and weather station data and found out that wind products in all of those have low correlation i.e less than 0.2. To investigate we compared Jra55 and NCEP to a shipboard data gathered during 2014 in time span of 2 months in Arctic and found that JRA55 had -0.20 correlation, RMSE of 7.36 and bias of -1.3, NCEP had correlation of 0.10, RMSE of 5.73 and bias of -1.40 when compared with high frequency data on

each cruise, reinforcing our suspicion of high frequency wind as a source of error in water currents."

Reviewer #2

R2-1)"The motivation of the paper is to estimate the mixed layer depth from the ocean model; however, the following sections do not seem to have any connections to the mixed layer depth estimation: 3.3.1 Geopotential height, 3.31 Vertical salinity temperature profiles and 3.6 Circulation and etc. This makes the interpretation of this paper difficult and results in many figures, which are not labeled correctly. For example, line 20 describes the sea-ice trajectory, but Fig. 10 contains color plot of geopotential and arrows, which are not described. Line 25 describes the geopotential, yet the referred figure 12 does not contain any information on the geopotential. I would keep the section 3.5 Mixed layer depth and omit 3.3.1, 3.6 and 3.7 unless there are clear connections to the change in the mixed layer depth."

AR) We agree with the reviewer that some sections needed to be removed and we also needed to state our hypothesis and aims more clearly. We edited the introduction to clearly state our target from the start and we removed the "geopotential height" and "general circulation" sections from manuscript. We also added a section where we calculate gas exchange on 3 grid points based on model outputs to demonstrate the impact of kept sections on gas exchange.

MC)Section "geopotential height" and "general circulation" are removed, Section 4 "Gas exchange estimation" is added. Page 2 Line 14 "Lacking sufficient data to constrain these processes, we wonder whether it is possible for a numerical model to adequately capture forcing of air-sea gas exchange in the sea ice zone and consequently improve predictions of air-sea flux. The parameters of interest are sea ice concentration (or fraction of open water), sea ice velocity, mixed layer depth, and water current speed and direction in the ice-ocean boundary layer (IOBL) (Loose et al. 2014). Here we use the budget of 222Rn gas in the IOBL as an example, because the radon-deficit

method has emerged as on of the principle methods to estimate k in ice-covered waters (Rutgers Van Der Loeff et al. 2014; Loose et al. 2016)."

R2-2)"The paper claims in the abstract "Overall, we find the course resolution model to be an inadequate surrogate for sparse data, however the simulation results are a slight improvement over several of the simplifying assumptions that are often made when surface ocean geochemistry, including the use of a constant mixed layer depth and a velocity profile that is purely wind-driven." I agree with the first claim "find the course resolution model to be an inadequate surrogate for sparse data"; however I do not see where the second claim, "the simulation results are a slight improvement over several of the simplifying assumptions that are often made when surface ocean geochemistry", is supported in the paper. The authors need to show the rate of air-sea gas exchange calculated from the mixed layer depth from 1) the ocean model results and 2) the conventional method, and discuss the difference in the estimates."

AR-MC) To demonstrate each parameter we calculated in our manuscript, we added section 4 Gas exchange estimation MC)Section : 4 - Gas exchange estimation and Figure 10 added.

R2-3)"I am concerned with the use of salt plume parameterization in the model configuration, A1. The A1 has relatively high vertical resolution of 2 m down to upper 50 m, whereas the salt plume parameterization is tested for models having the vertical resolution of _10 m. The salt plume parameterization is known to suppress the mixing to maintain a reasonable mixed layer depth in the polar regions. The parameterization is meant for a coarse vertical resolution model (_10 m). The vertical resolution in A1 is 5 times higher, and I am concerned that the parameterization in A1 is over suppressing the mixed layer depth. In fact, Figs 11, 13 20, 21 all points to the underestimation of the mixed layer depth compared to the observations. I suggest switching off the salt plume parameterization and evaluating the mixed layer depth in comparison to the existing model results and observations."

AR) After we noticed the reviewers concern with salt plume parameterization, we preformed a series of 1D-tests with and without the scheme and found no dependency of this scheme on vertical resolution. We also compared mixed layer from A3 (no salt plume) to A1 and A2 (salt plume) and confirmed that this parametrization is not responsible for mixed layer bias.

MC) Page 9 Line 24 "One last important note is the effect of the salt plume parameterization (SPP) on MLD. Nguyen et al. (2009) demonstrated the need to remove the artificial excessive vertical mixing in coarse horizontal resolution models. To rule out the dependency of this parametrization to vertical resolution as a source in MLD bias, we preformed a suit of 1D tests, with and without the SPP on variety of vertical resolutions (not shown here) and sea ice melting/freezing scenarios and confirmed that SPP is not dependent on vertical grid spacing. We also investigated MLD in A3 (no SPP) run compared to A2, and confirmed the average MLD is the same between these 2 runs."

Minor Comments

"Line 10-15: It is unclear if the lateral boundary conditions are prescribed from the climatology or seasonal output from ECCO2 and JRA25. It is also unclear why the authors use output from two different models ECCO2 and JRA25. Map of the domain showing bathymetry will be helpful."

AR) Thanks you. We changed the text to clarify that our boundary conditions are from ECCO2 and surface forcing on the domain is from JRA25.

MC)Page 4 Line 13 Surface forcings are from the 25 year Japanese Reanalysis Project (JRA25)(Onogi et al. 2007) for 36 km and 9 km runs and The European Center for Medium-Range Weather Forecasts (ECMWF) analysis for 2 km run. Page 4 Line 19 The horizontal boundary condition for the 36 km and 9 km configurations comes from existing global ECCO2 model outputs (Marshall et al. 1997; Menemenlis et al. 2008; Losch et al. 2010; Heimbach et al. 2010).

"Equation 3: What is the definition of i?"

MC)this section has been removed as part of R1-7.